# Near Optimal Sketching of Low-Rank Tensor Regression

**Jarvis Haupt**[1]
jdhaupt@umn.edu

**Xingguo Li**[1,2]
lixx1661@umn.edu

**David P. Woodruff**[3]
dwoodruf@cs.cmu.edu [*]

[1] University of Minnesota    [2]Georgia Tech    [3]Carnegie Mellon University

## Abstract

We study the least squares regression problem

$$\min_{\Theta \in \mathbb{R}^{p_1 \times \cdots \times p_D}} \|\mathcal{A}(\Theta) - b\|_2^2,$$

where $\Theta$ is a low-rank tensor, defined as $\Theta = \sum_{r=1}^{R} \theta_1^{(r)} \circ \cdots \circ \theta_D^{(r)}$, for vectors $\theta_d^{(r)} \in \mathbb{R}^{p_d}$ for all $r \in [R]$ and $d \in [D]$. Here, $\circ$ denotes the outer product of vectors, and $\mathcal{A}(\Theta)$ is a linear function on $\Theta$. This problem is motivated by the fact that the number of parameters in $\Theta$ is only $R \cdot \sum_{d=1}^{D} p_d$, which is significantly smaller than the $\prod_{d=1}^{D} p_d$ number of parameters in ordinary least squares regression. We consider the above CP decomposition model of tensors $\Theta$, as well as the Tucker decomposition. For both models we show how to apply data dimensionality reduction techniques based on *sparse* random projections $\Phi \in \mathbb{R}^{m \times n}$, with $m \ll n$, to reduce the problem to a much smaller problem $\min_{\Theta} \|\Phi\mathcal{A}(\Theta) - \Phi b\|_2^2$, for which $\|\Phi\mathcal{A}(\Theta) - \Phi b\|_2^2 = (1 \pm \varepsilon)\|\mathcal{A}(\Theta) - b\|_2^2$ holds simultaneously for all $\Theta$. We obtain a significantly smaller dimension and sparsity in the randomized linear mapping $\Phi$ than is possible for ordinary least squares regression. Finally, we give a number of numerical simulations supporting our theory.

## 1 Introduction

For a sequence of $D$-way design tensors $A_i \in \mathbb{R}^{p_1 \times \cdots \times p_D}$, $i \in [n] \triangleq \{1, \ldots, n\}$, suppose we observe noisy linear measurements of an unknown $D$-way tensor $\Theta \in \mathbb{R}^{p_1 \times \cdots \times p_D}$, given by

$$b = \mathcal{A}_i(\Theta) + z, \ \ b, z \in \mathbb{R}^n, \tag{1}$$

where $\mathcal{A}(\cdot) : \mathbb{R}^{p_1 \times \cdots \times p_D} \to \mathbb{R}^n$ is a linear function with $\mathcal{A}_i(\Theta) = \langle A_i, \Theta \rangle = \mathrm{vec}(A_i)^\top \mathrm{vec}(\Theta)$ for all $i \in [n]$, $\mathrm{vec}(X)$ is the vectorization of a tensor $X$, and $z = [z_1, \ldots, z_n]^\top$ corresponds to the observation noise. Given the design tensors $\{A_i\}_{i=1}^n$ and noisy observations $b = [b_1, \ldots, b_n]^\top$, a natural approach for estimating the parameter $\Theta$ is to use the *Ordinary Least Square* (OLS) estimation for the tensor regression problem, i.e., to solve

$$\min_{\Theta \in \mathbb{R}^{p_1 \times \cdots \times p_D}} \|\mathcal{A}(\Theta) - b\|_2^2. \tag{2}$$

Tensor regression has been widely studied in the literature. Applications include computer vision [8, 19, 34], data mining [5], multi-model ensembles [32], neuroimaging analysis [15, 36], multitask learning [21, 31], and multivariate spatial-temporal data analysis [1, 11]. In these applications, modeling the unknown parameters as a tensor is what is needed, as it allows for learning data that has multi-directional relations, such as in climate prediction [33], inherent structure learning with multi-dimensional indices [21], and hand movement trajectory decoding [34].

---

[*]The authors are listed in alphabetical order. Correspondence to: Xingguo Li <lixx1661@umn.edu>. The authors acknowledge support from University of Minnesota Startup Funding and Doctoral Dissertation Fellowship from University of Minnesota.

Due to the high dimensionality of tensor data, structured learning based on low-rank tensor decompositions, such as CANDECOMP/PARAFAC (CP) decomposition and Tucker decomposition models [13, 24], have been proposed in order to obtain tractable tensor regression problems. As discussed more below, requiring the unknown tensor to be low-rank significantly reduces the number of unknown parameters.

We consider low-rank tensor regression problems based on the CP decomposition and Tucker decomposition models. For simplicity, we first focus on the CP model, and later extend our analysis to the Tucker model. Suppose that $\Theta$ admits a rank-$R$ CP decomposition, that is,

$$\Theta = \sum_{r=1}^{R} \theta_1^{(r)} \circ \cdots \circ \theta_D^{(r)}, \tag{3}$$

where $\theta_d^{(r)} \in \mathbb{R}^{p_d}$ for all $r \in [R]$, $d \in [D]$, and $\circ$ is the outer product of vectors. For convenience, we reparameterize the set of low-rank tensors by its matrix slabs/factors:

$$\mathcal{S}_{D,R} \triangleq \left\{ [[\Theta_1, \ldots, \Theta_D]] \mid \Theta_d = [\theta_d^{(1)}, \ldots, \theta_d^{(R)}] \in \mathbb{R}^{p_d \times R}, \text{ for all } d \in [D] \right\}.$$

Then we can rewrite model (1) in a compact form

$$b = A(\Theta_D \odot \cdots \odot \Theta_1) 1_R + z, \tag{4}$$

where $A = [\mathrm{vec}(A_1), \cdots, \mathrm{vec}(A_n)]^\top \in \mathbb{R}^{n \times \prod_{d=1}^{D} p_d}$ is the matricization of all design tensors, $1_R = [1, \ldots, 1] \in \mathbb{R}^R$ is a vector of all 1s, $\otimes$ is the Kronecker product, and $\odot$ is the Khatri-Rao product. In addition, the OLS estimation for tensor regression (2) can be rewritten as the following nonconvex problem in terms of low-rank tensor parameters $[[\Theta_1, \ldots, \Theta_D]]$,

$$\min_{\vartheta \in \mathcal{S}_{\odot D, R}} \|A\vartheta - b\|_2^2, \quad \text{where} \tag{5}$$

$$\mathcal{S}_{\odot D, R} \triangleq \left\{ (\Theta_D \odot \cdots \odot \Theta_1) 1_R \in \mathbb{R}^{\prod_{d=1}^{D} p_d} \mid [[\Theta_1, \ldots, \Theta_D]] \in \mathcal{S}_{D,R} \right\}.$$

The number of parameters for a general tensor $\Theta \in \mathbb{R}^{p_1 \times \cdots \times p_D}$ is $\prod_{d=1}^{D} p_d$, which may be prohibitive for estimation even for small values of $\{p_d\}_{d=1}^{D}$. The benefit of the low-rank tensor model (3) is that it dramatically reduces the degrees of freedom of the unknown tensor from $\prod_{d=1}^{D} p_d$ to $R \cdot \sum_{d=1}^{D} p_d$, where we are typically interested in the case when $R \le p_d$ for all $d \in [D]$. For example, a typical MRI image has size $256^3 \approx 1.7 \times 10^7$, while using the low-rank model with $R = 10$, we reduce the number of unknown parameters to $256 \times 3 \times 10 \approx 8 \times 10^3 \ll 10^7$. This significantly increases the applicability of the tensor regression model in practice.

Nevertheless, solving the tensor regression problem (5) is still expensive in terms of both computation and memory requirements, for typical settings, when $n \gg R \cdot \sum_{d=1}^{D} p_d$. In particular, the per iteration complexity is at least linear in $n$ for popular algorithms such as block alternating minimization and block gradient descent [27, 28]. In addition, in order to store $A$, it takes $n \cdot \prod_{d=1}^{D} p_d$ words of memory. Both of these aspects are undesirable when $n$ is large. This motivates us to consider data dimensionality reduction techniques, also called *sketching*, for the tensor regression problem.

Instead of solving (5), we consider the simple *Sketched Ordinary Least Square* (SOLS) problem:

$$\min_{\vartheta \in \mathcal{S}_{\odot D, R}} \|\Phi A\vartheta - \Phi b\|_2^2, \tag{6}$$

where $\Phi \in \mathbb{R}^{m \times n}$ is a random matrix (specified in Section 2). Importantly, $\Phi$ will satisfy two properties, namely (1) $m \ll n$ so that we significantly reduce the size of the problem, and (2) $\Phi$ will be very sparse so that $\Phi v$ can be computed very quickly for any $v \in \mathbb{R}^n$.

Naïvely applying existing analyses of sketching techniques for least squares regression requires $m = \Omega(\prod_{d=1}^{D} p_d)$, which is prohibitive (for a survey, see, e.g., [30]). In this paper, our main contribution is to show that it is possible to use a sparse Johnson-Lindenstrauss transformation as our sketching matrix for the CP model of low-rank tensor regression, with constant column sparsity and dimension $m = R \cdot \sum_{d=1}^{D} p_d$, up to poly-logarithmic (polylog) factors. Note that our dimension matches the number of intrinsic parameters in the CP model. Further, we stress that we *do not* assume anything about the tensor, such as orthogonal matrix slabs/factor, or incoherence; our dimensionality

reduction works for *arbitrary* tensors. We show, with the above sparsity and dimenion, that with constant probability, simultaneously for all $\vartheta \in \mathcal{S}_{\odot,D,R}$,

$$\|\Phi A\vartheta - \Phi b\|_2^2 = (1 \pm \varepsilon)\|A\vartheta - b\|_2^2.$$

This implies that any solution to (6) has the same cost as in (5) up to a $(1 + \varepsilon)$-factor. In particular, by solving (6) we obtain a $(1 + \varepsilon)$-approximation to (5). We note that our dimensionality reduction technique is not tied to any particular algorithm; that is, if one runs any algorithm or heuristic on the reduced (sketched) problem, obtaining an $\alpha$-approximate solution $\vartheta$, then $\vartheta$ is also a $(1+\epsilon)\alpha$-approximate solution to the original problem. Our result is the first non-trivial dimensionality reduction for this problem, i.e., dimensionality reduction better than $\prod_{d=1}^{D} p_d$, which is trivial by ignoring the low-rank structure of the tensor, and which achieves a relative error $(1 + \varepsilon)$-approximation.

While it may be possible to apply dimensionality reduction methods directly in alternating minimization methods for solving tensor regression, unlike our method, such methods do not have provable guarantees and it is not clear how errors propagate across iterations. However, since we reduce the original problem to a smaller version of itself with a provable guarantee, one could further apply dimensionality reduction techniques as heuristics for alternating minimization on the smaller problem.

Our proof is based on a careful characterization of Talagrand's functional for the parameter space of low-rank tensors, providing a highly nontrivial analysis for what we consider to be a simple and practical algorithm. One of the main difficulties is dealing with general, non-orthogonal tensors, for which we are able to provide a careful re-parameterization in order to bound the so-called Finsler metric; interestingly, for non-orthogonal tensors it is always possible to *partially* orthogonalize them, and this partial orthogonalization turns out to suffice for our analysis. We give precise details below. We also provide numerical evaluations on both synthetic and real data to demonstrate the empirical performance of our algorithm.

**Notation**. For scalars $x, y \in \mathbb{R}$, let $x = (1 \pm \varepsilon)y$ if $x \in [(1 - \varepsilon)y, (1 + \varepsilon)y]$, $x \lesssim (\gtrsim)y$ if $x \leq (\geq)c_1 y$, $\text{poly}(x) = x^{c_2}$ and $\text{polylog}(x, y) = (\log x)^{c_3} \cdot (\log y)^{c_4}$ for some universal constants $c_1, c_2, c_3, c_4 > 0$. We also use standard asymptotic notations $\mathcal{O}(\cdot)$ and $\Omega(\cdot)$. Given a matrix $A \in \mathbb{R}^{m \times n}$, we denote $\|A\|_2$ as the spectral norm, $\text{span}(A) \subseteq \mathbb{R}^m$ as the subspace spanned by the columns of $A$, $\sigma_{\max}(A)$ and $\sigma_{\min}(A)$ as the largest and smallest singular values of $A$, respectively, and $\kappa_A = \sigma_{\max}(A)/\sigma_{\min}(A)$ as the condition number. We use $\text{nnz}(A)$ to denote the number of nonzero entries of $A$, and $\mathcal{P}_A$ as the projection operator onto $\text{span}(A)$. Given two matrices $A = [a_1, \ldots, a_n] \in \mathbb{R}^{m \times n}$ and $B = [b_1, \ldots, b_q] \in \mathbb{R}^{p \times q}$, $A \otimes B = [a_1 \otimes B, \ldots, a_n \otimes B] \in \mathbb{R}^{mp \times nq}$ denotes the Kronecker product, and $A \odot B = [a_1 \otimes b_1, \ldots, a_n \otimes b_n] \in \mathbb{R}^{mp \times n}$ denotes the Khatri-Rao product with $n = q$. We let $\mathcal{B}_n \subset \mathbb{R}^n$ be the unit sphere in $\mathbb{R}^n$, i.e., $\mathcal{B}_n = \{x \in \mathbb{R}^n \mid \|x\|_2 = 1\}$, $\mathbb{P}(\cdot)$ be the probability of an event, and $\mathbb{E}(\cdot)$ denotes the expectation of a random variable. Without further specification, we denote $\prod = \prod_{d=1}^{D}$ and $\sum = \sum_{d=1}^{D}$. We further summarize the dimension parameters for ease of reference. Given a tensor $\Theta$, $D$ is the number of ways, $p_d$ is the dimension of the $d$-th way for $d \in [D]$. $R$ is the rank of $\Theta$ for all ways under the CP decomposition, and $R_d$ is the rank of the $d$-th way under the Tucker decomposition for $d \in [D]$. $n$ is the number of observations for tensor regression. $m$ is the sketching dimension and $s$ is the sparsity of each column in a sparse Johnson-Lindenstrauss transformation.

## 2  Background

We start with a few important definitions.

**Definition 1** (Oblivious Subspace Embedding)**.** Suppose $\Pi$ is a distribution on $m \times n$ matrices where $m$ is a function of parameters $n, d$, and $\varepsilon$. Further, suppose that with probability at least $1 - \delta$, for any fixed $n \times d$ matrix $A$, a matrix $\Phi$ drawn from $\Pi$ has the property that $\|\Phi Ax\|_2^2 = (1 \pm \varepsilon)\|Ax\|_2^2$ simultaneously for all $x \in \mathcal{X} \subseteq \mathbb{R}^d$. Then $\Pi$ is an $(\varepsilon, \delta)$ *oblivious subspace embedding* (OSE) of $\mathcal{X}$.

An OSE $\Phi$ preserves the norm of vectors in a certain set $\mathcal{X}$ after linear transformation by $A$. This is widely studied as a key property for sketching based analyses (see [30] and the references therein). We want to show an analogous property when $\mathcal{X}$ is parameterized by low-rank tensors.

**Definition 2** (Leverage Scores)**.** Given $A \in \mathbb{R}^{n \times d}$, let $Z \in \mathbb{R}^{n \times d}$ have orthonormal columns that span the column space of $A$. Then $\ell_i^2(A) = \|e_i^\top Z\|_2^2$ is the $i$-th *leverage score* of $A$.

Leverage scores play an important role in randomized matrix algorithms [7, 16, 17]. Calculating the leverage scores naïvely by orthogonalizing $A$ requires $\mathcal{O}(nd^2)$ time. It is shown in [3] that the leverage scores of $A$ can be approximated individually up to a constant multiplicative factor in $\mathcal{O}(\text{nnz}(A)\log n + \text{poly}(d))$ time using sparse subspace embeddings. In our analysis, there will be a very mild dependence on the maximum leverage score of $A$ and the sparsity for the sketching matrix $\Phi$. Note that we do not need to calculate the leverage scores.

**Definition 3** (Talagrand's Functional). Given a (semi-)metric $\rho$ on $\mathbb{R}^n$ and a bounded set $\mathcal{S} \subset \mathbb{R}^n$, *Talagrand's $\gamma_2$-functional* is

$$\gamma_2(S, \rho) = \inf_{\{\mathcal{S}_r\}_{r=0}^{\infty}} \sup_{x \in S} \sum_{r=0}^{\infty} 2^{r/2} \cdot \rho(x, \mathcal{S}_r), \tag{7}$$

where $\rho(x, \mathcal{S}_r)$ is a distance from $x$ to $\mathcal{S}_r$ and the infimum is taken over all collections $\{\mathcal{S}_r\}_{r=0}^{\infty}$ such that $\mathcal{S}_0 \subset \mathcal{S}_1 \subset \ldots \subset \mathcal{S}$ with $|\mathcal{S}_0| = 1$ and $|\mathcal{S}_r| \leq 2^{2^r}$.

A closely related notion of the $\gamma_2$-functional is the *Gausssian mean width*: $\mathcal{G}(\mathcal{S}) = \mathbb{E}_g \sup_{x \in \mathcal{S}} \langle g, x \rangle$, where $g \sim \mathcal{N}_n(0, I_n)$. For any bounded $\mathcal{S} \subset \mathbb{R}^n$, $\mathcal{G}(\mathcal{S})$ and $\gamma_2(\mathcal{S}, \| \cdot \|_2)$ differ multiplicatively by at most a universal constant in Euclidean space [25]. Finding a tight upper bound on the $\gamma_2$-functional for the parameter space of low-rank tensors is key to our analysis.

**Definition 4** (Finsler Metric). Let $E, E' \subset \mathbb{R}^n$ be $p$-dimensional subspaces. The *Finsler metric* of $E$ and $E'$ is $\rho_{\text{Fin}}(E, E') = \| \mathcal{P}_E - \mathcal{P}_{E'} \|_2$, where $\mathcal{P}_E$ is the projection onto the subspace $E$.

The Finsler metric is the semi-metric used in the $\gamma_2$-functional in our analysis. Note that $\rho_{\text{Fin}}(E, E') \leq 1$ always holds for any $E$ and $E'$ [23].

**Definition 5** (Sparse Johnson-Lindenstrauss Transforms). Let $\sigma_{ij}$ be independent Rademacher random variables, i.e., $\mathbb{P}(\sigma_{ij} = 1) = \mathbb{P}(\sigma_{ij} = -1) = 1/2$, and let $\delta_{ij} : \Omega_\delta \to \{0, 1\}$ be random variables, independent of the $\sigma_{ij}$, with the following properties:

(i) $\delta_{ij}$ are negatively correlated for fixed $j$, i.e., for all $1 \leq i_1 < \ldots < i_k \leq m$, we have $\mathbb{E}\left(\prod_{t=1}^{k} \delta_{i_t, j}\right) \leq \prod_{t=1}^{k} \mathbb{E}(\delta_{i_t, j}) = \left(\frac{s}{m}\right)^k$;

(ii) There are $s = \sum_{i=1}^{m} \delta_{ij}$ nonzero $\delta_{ij}$ for a fixed $j$; and

(iii) The vectors $(\delta_{ij})_{i=1}^{m}$ are independent across $j \in [n]$.

Then $\Phi \in \mathbb{R}^{m \times n}$ is a *sparse Johnson-Lindenstrauss transform* (SJLT) matrix if $\Phi_{ij} = \frac{1}{\sqrt{s}} \sigma_{ij} \delta_{ij}$.

The SJLT has several benefits [4, 12, 30]. First, the computation of $\Phi x$ takes only $\mathcal{O}(\text{nnz}(x))$ time when $s$ is a constant. Second, storing $\Phi$ takes only $sn$ memory instead of $mn$, which is significant when $s \ll m$. This can often further be reduced by drawing the entries of $\Phi$ from a limited independent family of random variables.

We will use an SJLT matrix as the sketching matrix $\Phi$ in our analysis and our goal will be to show sufficient conditions on the sketching dimension $m$ and per-column sparsity $s$ such that the analogue of the OSE property holds for low-rank tensor regression. Specifically, we provide sufficient conditions for the SJLT matrix $\Phi \in \mathbb{R}^{m \times n}$ to preserve the cost of all solutions for tensor regression, i.e., bounds on $m$ and $s$ for which

$$\mathbb{E}_\Phi \sup_{x \in \mathcal{T}} \left| \|\Phi x\|_2^2 - 1 \right| < \frac{\varepsilon}{10}, \tag{8}$$

where $\varepsilon$ is a given precision and $\mathcal{T}$ is a normalized space parameterized as the union of certain subspaces of $A$, which will be further discussed in the following sections. Note that by linearity, it is sufficient to consider $x$ with $\|x\|_2 = 1$ in the above, which explains the form of (8). Moreover, by Markov's inequality, (8) implies that simultaneously for all $\vartheta = \text{vec}(\Theta) \in \mathcal{S}_{\odot D, R}$, where $\Theta$ admits a low-rank tensor decomposition, with probability at least 9/10, we have

$$\|\Phi A \vartheta - \Phi b\|_2^2 = (1 \pm \varepsilon) \|A \vartheta - b\|_2^2, \tag{9}$$

which allows us to minimize the much smaller sketched problem to obtain parameters $\vartheta$ which, when plugged into the original objective function, provide a multiplicative $(1 + \varepsilon)$-approximation.

# 3 Dimensionality Reduction for CP Decomposition

We start with the following notation. Given a tensor $\Theta = \sum_{r=1}^{R} \theta_1^{(r)} \circ \cdots \circ \theta_D^{(r)}$, where $\theta_d^{(r)} \in \mathbb{R}^{p_d}$ for all $d \in [D]$ and $r \in [R]$, we fix all but $\theta_1^{(r)}$ for $r \in [R]$, and denote

$$A^{\left\{\theta_{\backslash 1}^{(r)}\right\}} = \left[A^{\theta_{\backslash 1}^{(1)}}, \ldots, A^{\theta_{\backslash 1}^{(R)}}\right] \in \mathbb{R}^{n \times R p_1},$$

where $A^{\theta_{\backslash 1}^{(i)}} = \sum_{j_D=1}^{p_D} \cdots \sum_{j_2=1}^{p_2} A^{(j_D,\ldots,j_2)} \theta_{D,j_D}^{(i)} \cdots \theta_{2,j_2}^{(i)}$, $\theta_{d,j_d}^{(i)}$ is the $j_d$-th entry of $\theta_d^{(i)}$, and $A^{(j_D,\ldots,j_2)} \in \mathbb{R}^{n \times p_1}$ is a column submatrix of $A$ indexed by $j_D \in [p_D], \ldots, j_2 \in [p_2]$, i.e., $A = \left[A^{(1,\ldots,1)}, \ldots, A^{(p_D,\ldots,p_2)}\right] \in \mathbb{R}^{n \times \prod p_d}$. The above parameterization allows us to view tensor regression as preserving the norms of vectors in an infinite union of subspaces, described in more detail in the full version of our paper [10]. Then we rewrite the observation model (4) as

$$b = A \cdot \sum_{r=1}^{R} \theta_D^{(r)} \otimes \cdots \otimes \theta_1^{(r)} + z = \sum_{r=1}^{R} A^{\theta_{\backslash 1}^{(r)}} \cdot \theta_1^{(r)} + z = A^{\left\{\theta_{\backslash 1}^{(r)}\right\}} \cdot \left[\theta_1^{(1)\top} \ldots \theta_1^{(R)\top}\right]^{\top} + z.$$

## 3.1 Main Result

The parameter space for the tensor regression problem (1) is a subspace of $\mathbb{R}^{\prod p_d}$, i.e., $\mathcal{S}_{\odot D,R} \subset \mathbb{R}^{\prod p_d}$. Therefore, a naïve application of sketching requires $m \gtrsim \prod p_d / \varepsilon^2$ in order for (9) to hold [18]. The following theorem provides sufficient conditions to guarantee $(1 + \varepsilon)$-approximation of the objective for low-rank tensor regression under the CP decomposition model.

**Theorem 1.** Suppose $R \leq \max_d p_d / 2$ and $\max_{i \in [n]} \ell_i^2(A) \leq 1/(R \sum_{d=2}^{D} p_d)^2$. Let

$$\mathcal{T} = \bigcup_{r \in [R], d \in [D]} \left\{\frac{A\vartheta - A\varphi}{\|A\vartheta - A\varphi\|_2} \Big| \vartheta = \sum_{r=1}^{R} \theta_D^{(r)} \otimes \cdots \otimes \theta_1^{(r)}, \varphi = \sum_{r=1}^{R} \phi_D^{(r)} \otimes \cdots \otimes \phi_1^{(r)}, \theta_d^{(r)}, \phi_d^{(r)} \in \mathcal{B}_{p_d}\right\}$$

and let $\Phi \in \mathbb{R}^{m \times n}$ be an SJLT matrix with column sparsity $s$. Then with probability at least $9/10$, (9) holds if $m$ and $s$ satisfy, respectively,

$$m \gtrsim R \sum p_d \log \left(DR\kappa_A \sum p_d\right) \text{polylog}(m, n)/\varepsilon^2 \text{ and } s \gtrsim \log^2 \left(\sum p_d\right) \text{polylog}(m, n)/\varepsilon^2.$$

From Theorem 1, we have that for an SJLT matrix $\Phi \in \mathbb{R}^{m \times n}$ with $m = \Omega(R \sum p_d)$ and $s = \Omega(1)$, up to logarithmic factors, we can guarantee $(1 + \varepsilon)$-approximation of the objective. The sketching complexity of $m$ is nearly optimal compared with the number of free parameters for the CP decomposition model, i.e., $R(\sum p_d - D + 1)$, up to logarithmic factors. Here wo do not make any orthogonality assumption on the tensor factors $\theta_d^{(r)}$, and show in our analysis that the general tensor space $\mathcal{T}$ can be paramterized in terms of an orthogonal one if $R \leq \max_d p_d / 2$ holds. The condition $R \leq \max_d p_d / 2$ is not restrictive in our setting, as we are interested in low-rank tensors with $R \leq p_d$. Note that we achieve a $(1 + \varepsilon)$-approximation in objective function value for arbitrary tensors; if one wants to achieve closeness of the underlying parameters one needs to impose further assumptions on the model, such as the form of the noise distribution or structural properties of $A$ [20, 36].

Our maximum leverage score assumption is very mild and much weaker than the standard incoherence assumptions used for example, in matrix completion, which allow for uniform sampling based approaches. For example, our assumption states that the maximum leverage score is at most $1/(R \sum_{d=2}^{D} p_d)^2$. In the typical overconstrained case, $n \gg \prod p_d$, and in order for uniform sampling to provide a subspace embedding, one needs the maximum leverage score to be at most $R \sum p_d / n$ (see, e.g., Section 2.4 of [30]), which is much less than $1/(R \sum_{d=2}^{D} p_d)^2$ when $n$ is large, and so uniform sampling fails in our setting. Moreover, it is also possible to apply a standard idea to flatten the leverage scores of a deterministic design $A$ based on the Subsampled Randomized Hadamard Transformation (SRHT) using the Walsh-Hadamard matrix [9, 26]. Note that applying the SRHT to an $n \times d$ matrix $A$ only takes $O(nd \log n)$ time, which if $A$ is dense, is the same amount of time one needs just to read $A$ (up to a $\log n$ factor). Further details are deferred to the full version of our paper [10].

## 3.2 Proof Sketch of Our Analysis for a Basic Case

We provide a sketch of our analysis for the case when $R = 1$ and $D = 2$, i.e., $\Theta$ is rank 1 matrix. The analysis for more general cases is more involved, but with similar intuition. Details of the analyses are deferred to the full version of our paper, where we start with a proof for the most basic cases and gradually build up the proof for the most general case.

Let $A^v = \sum_{i=1}^{p_2} A^{(i)} v_i$, where $A = [A^{(1)}, \dots, A^{(p_2)}] \in \mathbb{R}^{n \times p_2 p_1}$ with $A^{(i)} \in \mathbb{R}^{n \times p_1}$ for all $i \in [p_2]$, $\mathcal{V} = \bigcup_{\widetilde{\mathcal{W}}} \{\text{span}[A^{v_1}, A^{v_2}]\}$, and $\widetilde{\mathcal{W}} = \{v_1, v_2 \in \mathcal{B}_{p_2} \text{ with } \langle v_1, v_2 \rangle = 0\}$. We start with an illustration that the set $\mathcal{T}$ can be reparameterized to the following set with respect to tensors with orthogonal factors:

$$\mathcal{T} = \bigcup_{E \in \mathcal{V}} \{x \in E \mid \|x\|_2 = 1\}.$$

Suppose $\langle v_1, v_2 \rangle \neq 0$. Let $v_2 = \alpha v_1 + \beta z$ for some $\alpha, \beta \in \mathbb{R}$ and a unit vector $z \in \mathbb{R}^{p_2}$, where $\langle v_1, z \rangle = 0$. Then we have

$$\frac{Ax - Ay}{\|Ax - Ay\|_2} = \frac{A^{v_1} u_1 - A^{v_2} u_2}{\|A^{v_1} u_1 - A^{v_2} u_2\|_2} = \frac{A^{v_1}(u_1 - \alpha u_2) - A^z(\beta u_2)}{\|A^{v_1}(u_1 - \alpha u_2) - A^z(\beta u_2)\|_2},$$

which is equivalent to $\langle v_1, v_2 \rangle = 0$ by reparameterizing $z$ as $v_2$.

Based on known dimensionality reduction results [2, 6] (see further details in the full version [10]), the main quantities needed for bounding properties of $\Phi$ are the quantities $p_{\mathcal{V}}$, $\gamma_2^2(\mathcal{V}, \rho_{\text{Fin}})$, $\mathcal{N}(\mathcal{V}, \rho_{\text{Fin}}, \varepsilon_0)$, and $\int_0^{\varepsilon_0} (\log \mathcal{N}(\mathcal{V}, \rho_{\text{Fin}}, t))^{1/2} dt$, where $\mathcal{N}(\mathcal{V}, \rho_{\text{Fin}}, t)$ is the covering number of $\mathcal{V}$ under the Finsler metric using balls of radius $t$ and $p_{\mathcal{V}} = \sup_{v_1, v_2 \in \mathcal{B}_{p_2}, \langle v_1, v_2 \rangle = 0} \dim \{\text{span}(A^{v_1, v_2})\} \leq 2p_1$. Bounding these quantities for the space of low-rank tensors is new and is our main technical contribution. These will be addressed separately as follows.

**Part 1: Bound $p_{\mathcal{V}}$.** Let $A^{v_1, v_2} = [A^{v_1}, A^{v_2}]$. It is straightforward that $p_{\mathcal{V}} \leq 2p_1$.

**Part 2: Bound $\gamma_2^2(\mathcal{V}, \rho_{\textbf{Fin}})$.** By the definition of $\gamma_2$-functional in (7) for the Finsler metric, we have

$$\gamma_2(\mathcal{V}, \rho_{\text{Fin}}) = \inf_{\{\overline{\mathcal{V}}_k\}_{k=0}^\infty} \sup_{A^{v_1, v_2} \in \mathcal{V}} \sum_{k=0}^\infty 2^{k/2} \cdot \rho_{\text{Fin}}(A^{v_1, v_2}, \overline{\mathcal{V}}_k),$$

where $\overline{\mathcal{V}}_k$ is an $\varepsilon_k$-net of $\mathcal{V}$, i.e., for any $A^{v_1, v_2} \in \mathcal{V}$ there exist $\overline{v}_1, \overline{v}_2 \in \mathcal{B}_{p_2}$ with $\langle \overline{v}_1, \overline{v}_2 \rangle = 0$, $\|v_1 - \overline{v}_1\|_2 \leq \eta_k$, and $\|v_2 - \overline{v}_2\|_2 \leq \eta_k$, such that $A^{\overline{v}_1, \overline{v}_2} \in \overline{\mathcal{V}}_k$ and $\rho_{\text{Fin}}(A^{v_1, v_2}, A^{\overline{v}_1, \overline{v}_2}) \leq \varepsilon_k$.

From Lemma 6, we have $\rho_{\text{Fin}}(A^{v_1, v_2}, \overline{\mathcal{V}}_k) \leq 2\kappa_A \eta_k$ for $\|v_1 - \overline{v}_1\|_2 \leq \eta_k$ and $\|v_2 - \overline{v}_2\|_2 \leq \eta_k$. On the other hand, we have that $\rho_{\text{Fin}}(A^{v_1, v_2}, \overline{\mathcal{V}}_k) \leq 1$ always holds. Therefore, we have $\rho_{\text{Fin}}(A^{v_1, v_2}, \overline{\mathcal{V}}_k) \leq \min\{2\kappa_A \eta_k, 1\}$. Let $k'$ be the smallest integer such that $2\kappa_A \eta_{k'} \leq 1$. Then

$$\gamma_2(\mathcal{V}, \rho_{\text{Fin}}) \leq \sum_{k=0}^\infty 2^{k/2} \rho_{\text{Fin}}(A^{v_1, v_2}, \overline{\mathcal{V}}_k) \leq \sum_{k=0}^{k'} 2^{k/2} + \sum_{k=k'+1}^\infty 2^{k/2} \rho_{\text{Fin}}(A^{v_1, v_2}, \overline{\mathcal{V}}_k). \tag{10}$$

Starting from $\eta_0 = 1$ and $|\overline{\mathcal{V}}_0| = 1$, for $k \geq 1$, we have $\eta_k < 1$ and $|\overline{\mathcal{V}}_k| \leq (3/\eta_k)^{p_2}$ [29]. Also from the $\gamma_2$-functional, we require $|\overline{\mathcal{V}}_k| \leq 2^{2^k} \leq (3/\eta_k)^{p_2}$, which implies

$$\sum_{k=0}^{k'} 2^{k/2} = \frac{2^{k'/2}}{\sqrt{2} - 1} \lesssim \sqrt{p_2 \log \frac{1}{\eta_{k'}}}. \tag{11}$$

For $k > k'$, we choose $\eta_{k+1} = \eta_k^2$ such that $(3/\eta_{k+1})^{p_2} \leq 2^{2^{k+1}}$. Then we have $|\overline{\mathcal{V}}_{k+1}| \leq 2^{2^{k+1}}$. By choosing $k'$ to be the smallest integer such that $(3/\eta_{k'+1})^{p_2} \leq 2^{2^{k'+1}}$ holds, we have

$$\sum_{k=k'+1}^\infty 2^{k/2} \cdot \rho_{\text{Fin}}(A^{v_1, v_2}, \overline{\mathcal{V}}_k) = 2^{k'/2} \cdot \sum_{t=1}^\infty 2^{t/2} \cdot \left(\frac{1}{2}\right)^{2^t} \leq 2^{k'/2} \lesssim \sqrt{p_2 \log \frac{1}{\eta_{k'}}}. \tag{12}$$

Combining (10) – (12), and choosing a small enough $\varepsilon_0$ such that $\varepsilon_0 \leq 2\kappa_A \eta_{k'}$, we have

$$\gamma_2^2(\mathcal{V}, \rho_{\mathrm{Fin}}) \lesssim p_2 \log \frac{\kappa_A}{\varepsilon_0}.$$

**Part 3: Bound $\mathcal{N}(\mathcal{V}, \rho_{\mathbf{Fin}}, \varepsilon_0)$ and $\int_0^{\varepsilon_0} [\log \mathcal{N}(\mathcal{V}, \rho_{\mathbf{Fin}}, t)]^{1/2} dt$.** From our choice from Part 2, $\varepsilon_0 \in (0,1)$ is a constant. Then it is straightforward that $\mathcal{N}(\mathcal{V}, \rho_{\mathrm{Fin}}, \varepsilon_0) \leq \left(\frac{3}{\varepsilon_0}\right)^{2p_2}$. From direct integration, this implies

$$\int_0^{\varepsilon_0} [\log \mathcal{N}(\mathcal{V}, \rho_{\mathrm{Fin}}, t)]^{1/2} dt \lesssim \varepsilon_0 \sqrt{p_2 \log \frac{1}{\varepsilon_0}}.$$

Combining the results in Parts 1, 2, and 3, we have that (9) holds if $m$ and $s$ satisfy, respectively

$$m \gtrsim \frac{\left(p_2 \log \frac{\kappa_A}{\varepsilon_0} + p_1 + p_2 \log \frac{1}{\varepsilon_0}\right) \cdot \mathrm{polylog}(m,n)}{\varepsilon^2} \quad \text{and}$$

$$s \gtrsim \frac{\left(\log^2 \frac{1}{\varepsilon_0} + \varepsilon_0^2 (p_1 + p_2) \log \frac{1}{\varepsilon_0}\right) \cdot \mathrm{polylog}(m,n)}{\varepsilon^2}.$$

We finish the proof by taking $\varepsilon_0 = 1/(p_1 + p_2)$.

## 4   Dimensionality Reduction for Tucker Decomposition

We start with a formal model description. Suppose $\Theta$ admits the following Tucker decomposition:

$$\Theta = \sum_{r_1=1}^{R_1} \cdots \sum_{r_D=1}^{R_D} G(r_1, \ldots, r_D) \cdot \theta_1^{(r_1)} \circ \cdots \circ \theta_D^{(r_D)}, \tag{13}$$

where $G \in \mathbb{R}^{R_1 \times \cdots \times R_D}$ is the core tensor and $\theta_d^{(r_d)} \in \mathbb{R}^{p_d}$ for all $r_d \in [R_d]$ and $d \in [D]$. Let

$A^{\theta_{\backslash 1}^{(r_1,\ldots,r_D)}} = \sum_{j_D=1}^{p_D} \cdots \sum_{j_2=1}^{p_2} A^{(j_D,\ldots,j_2)} \theta_{D,j_D}^{(r_D)} \cdots \theta_{2,j_2}^{(r_2)}$ and

$A^{\left\{\theta_{\backslash 1}^{\{r_d\}}\right\}} = \left[\sum_{r_2=1}^{R_2} \cdots \sum_{r_D=1}^{R_D} A^{\theta_{\backslash 1}^{(r_1,\ldots,r_D)}} G(1, r_2, \ldots, r_D), \ldots, \sum_{r_2=1}^{R_2} \cdots \sum_{r_D=1}^{R_D} A^{\theta_{\backslash 1}^{(r_1,\ldots,r_D)}} G(R_1, r_2, \ldots, r_D)\right].$

Then the observation model (4) can be written as

$b = A \sum_{r_1=1}^{R_1} \cdots \sum_{r_D=1}^{R_D} G(r_1, \ldots, r_D) \theta_D^{(r_D)} \otimes \cdots \otimes \theta_1^{(r_1)} + z = A^{\left\{\theta_{\backslash 1}^{\{r_d\}}\right\}} \left[\theta_1^{(1)\top} \ldots \theta_1^{(R_1)\top}\right]^\top + z.$

The following theorem provides sufficient conditions to guarantee $(1 + \varepsilon)$-approximation of the objective function for low-rank tensor regression under the Tucker decomposition model.

**Theorem 2.** Suppose $\mathrm{nnz}(G) \leq \max_d p_d / 2$ and $\max_{i \in [n]} \ell_i^2(A) \leq 1/(\sum_{d=2}^D R_d p_d + \mathrm{nnz}(G))^2$.

$$\text{Let } \mathcal{T} = \bigcup_{r \in [R], d \in [D]} \left\{ \frac{A\vartheta - A\varphi}{\|A\vartheta - A\varphi\|_2} \middle| \vartheta = \sum_{r_1=1}^{R_1} \cdots \sum_{r_D=1}^{R_D} G_1(r_1, \ldots, r_D) \cdot \theta_D^{(r_D)} \otimes \cdots \otimes \theta_1^{(r_1)}, \right.$$

$$\left. \varphi = \sum_{r_1=1}^{R_1} \cdots \sum_{r_D=1}^{R_D} G_2(r_1, \ldots, r_D) \cdot \phi_D^{(r_D)} \otimes \cdots \otimes \phi_1^{(r_1)}, \; \theta_d^{(r_d)}, \phi_d^{(r_d)} \in \mathcal{B}_{p_d} \right\}$$

and $\Phi \in \mathbb{R}^{m \times n}$ be an SJLT matrix with column sparsity $s$. Then with probability at least $9/10$, (9) holds if $m$ and $s$ satisfy

$$m \gtrsim C_1 \cdot \log\left(C_1 D \kappa_A R_1 \sqrt{\mathrm{nnz}(G)}\right) \cdot \mathrm{polylog}(m,n)/\varepsilon^2 \quad \text{and} \quad s \gtrsim \log^2 C_1 \cdot \mathrm{polylog}(m,n)/\varepsilon^2,$$

where $C_1 = \sum R_d p_d + \mathrm{nnz}(G)$.

From Theorem 2, we have that using an SJLT matrix $\Phi$ with $m = \Omega(\sum R_d p_d + \mathrm{nnz}(G))$ and $s = \Omega(1)$, up to logarithmic factors, we can guarantee $(1+\varepsilon)$-approximation of the objective function.

The sketching complexity of $m$ is near optimal compared with the number of free parameters for the Tucker decomposition model, i.e., $\sum R_d p_d + \text{nnz}(G) - \sum R_d^2$, up to logarithmic factors. Note that $\text{nnz}(G) \leq \prod R_d$, and thus the condition that $\text{nnz}(G) \leq \max_d p_d/2$ can be more restrictive than $R \leq \max_d p_d/2$ in the CP model when $\text{nnz}(G) > R$. This is due to the fact that the Tucker model is more "expressive" than the CP model for a tensor of the same dimensions. For example, if $R_1 = \cdots = R_D = R$, then the CP model (3) can be viewed as special case of the Tucker model (13) by setting all off-diagonal entries of the core tensor $G$ to be 0. Moreover, the conditions and results in Theorem 2 are essentially of the same order as those in Theorem 1 when $\text{nnz}(G) = R$, which indicates the tightness of our analysis.

## 5  Experiments

We study the performance of sketching for tensor regression through numerical experiments over both synthetic and real data sets. For solving the OLS problem for tensor regression (2), we use a cyclic block-coordinate minimization algorithm based on a tensor toolbox [35]. Specifically, in a cyclic manner for all $d \in [D]$, we fix all but one $\Theta_d$ of $[[\Theta_1, \ldots, \Theta_D]] \in \mathcal{S}_{D,R}$ and minimize the resulting quadratic loss function (2) with respect to $\Theta_i$, until the decrease of the objective is smaller than a predefined threshold $\tau$. For SOLS, we use the same algorithm after multiplying $A$ and $b$ with an SJLT matrix $\Phi$. All results are run on a supercomputer due to the large scale of the data. Note that our result is not tied to any specific algorithm and we can use any algorithm that solves OLS for low-rank tensors for solving SOLS for low-rank tensors.

For synthetic data, we generate the low-rank tensor $\Theta$ as follows. For each $d \in [D]$, we generate $R$ random columns with $\mathcal{N}(0,1)$ entries to form *non-orthogonal* tensor factors $\Theta_d = [\theta_d^{(1)}, \ldots, \theta_d^{(R)}]$ of $[[\Theta_1, \ldots, \Theta_D]] \in \mathcal{S}_{D,R}$ independently. We also generate $R$ real scalars $\alpha_1, \ldots, \alpha_R$ uniformly and independently from $[1, 10]$. Then $\Theta$ is formed by $\Theta = \sum_{r=1}^{R} \alpha_r \theta_1^{(r)} \circ \cdots \circ \theta_D^{(r)}$. The $n$ tensor designs $\{A_i\}_{i=1}^n$ are generated independently with i.i.d. $\mathcal{N}(0,1)$ entries for 10% of the entries chosen uniformly at random, and the remaining entries are set to zero. We also generate the noise $z$ to have i.i.d. $\mathcal{N}(0, \sigma_z^2)$ entries, and the generation of the SJLT matrix $\Phi$ follows Definition 5. For both OLS and SOLS, we use random initializations for $\Theta$, i.e., $\Theta_d$ has i.i.d. $\mathcal{N}(0,1)$ entries for all $d \in [D]$.

We compare OLS and SOLS for low-rank tensor regression under both the noiseless and noisy scenarios. For the noiseless case, i.e., $\sigma_z = 0$, we choose $R = 3$, $p_1 = p_2 = p_3 = 100$, $m = 5 \times R(p_1 + p_2 + p_3) = 4500$, and $s = 200$. Different values of $n = 10^4, 10^5$, and $10^6$ are chosen to compare both statistical and computational performances of OLS and SOLS. For the noisy case, the settings of all parameters are identical to those in the noiseless case, except that $\sigma_z = 1$. We provide a plot of the scaled objective versus the number of iterations for some random trials in Figure 1. The scaled objective is set to be $\|A\vartheta_{\text{SOLS}}^t - b\|_2^2/n$ for SOLS and $\|A\vartheta_{\text{OLS}}^t - b\|_2^2/n$ for OLS, where

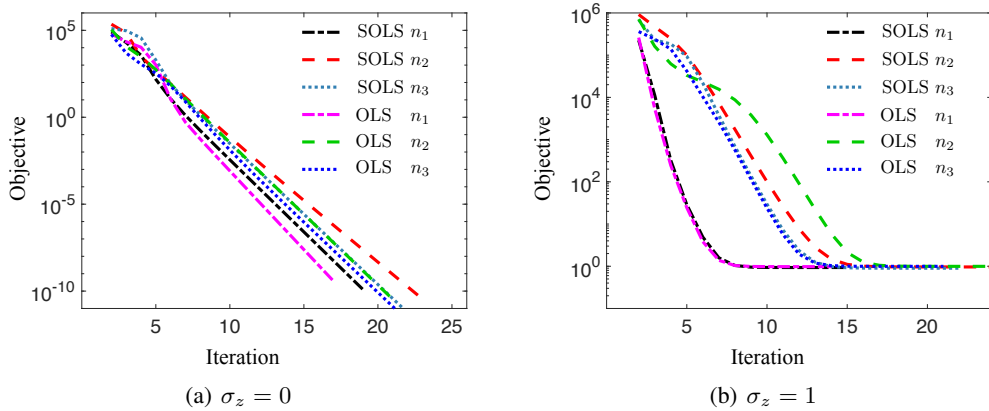

(a) $\sigma_z = 0$                          (b) $\sigma_z = 1$

Figure 1: Comparison of SOLS and OLS on synthetic data. The vertical axis corresponds to the scaled objectives $\|A\vartheta_{\text{SOLS}}^t - b\|_2^2/n$ for SOLS and $\|A\vartheta_{\text{OLS}}^t - b\|_2^2/n$ for OLS, where $\vartheta^t$ is the update in the $t$-th iteration. The horizontal axis corresponds to the number of iterations (passes of block-coordinate minimization for all blocks). For both the noiseless case $\sigma_z = 0$ and noisy case $\sigma_z = 1$, we set $n_1 = 10^4$, $n_2 = 10^5$, and $n_3 = 10^6$ respectively.

$\vartheta^t_{\text{SOLS}}$ and $\vartheta^t_{\text{OLS}}$ are the updates in the $t$-th iterations of SOLS and OLS respectively. Note the we are using $\|\Phi A\vartheta_{\text{SOLS}} - \Phi b\|_2^2/n$ as the objective function for solving the SOLS problem, but looking at the original objective $\|A\vartheta_{\text{SOLS}} - b\|_2^2/n$ for the solution of SOLS is ultimately what we are interested in. However, we have that the gap between $\|\Phi A\vartheta_{\text{SOLS}} - \Phi b\|_2^2/n$ and $\|A\vartheta_{\text{SOLS}} - b\|_2^2/n$ is very small in our results ($< 1\%$). The number of iterations is the number of passes of block-coordinate minimization for all blocks. We can see that OLS and SOLS require approximately the same number of iterations for comparable decrease in objective function value. However, since the SOLS instance has a much smaller size, its per iteration computational cost is much lower than that of OLS.

We further provide numerical results on the running time (CPU execution time) and the optimal scaled objectives in Table 1. Using the same stopping criterion, we see that SOLS and OLS achieve comparable objectives (within $< 5\%$ differences), matching our theory. In terms of the running time, SOLS is significantly faster than OLS, especially when $n$ is large compared to the sketching dimen sion $m$. For example, when $n = 10^6$, SOLS is more than 200 times faster than OLS while achieving a comparable objective function value with OLS. This matches with our theoretical results on the computational cost of OLS versus SOLS. Note that here we suppose that the rank is known for our simulation, which can be restrictive in practice. We observe that if we choose a moderately larger rank than the true rank of the underlying model, then the results are similar to what we discussed above. Smaller values of the rank result in a much deteriorated statistical performance for both OLS and SOLS.

We also examine sketching of low-rank tensor regression on a real dataset of MRI images [22]. The dataset consists of 56 frames of a human brain, each of which is of dimension $128 \times 128$ pixels, i.e., $p_1 = p_2 = 128$ and $p_3 = 56$. The generation of design tensors $\{A_i\}_{i=1}^n$ and linear measurements $b$ follows the same settings as for the synthetic data, with $\sigma_z = 0$. We choose three values of $R = 3, 5, 10$, and set $m = 5 \times R(p_1 + p_2 + p_3)$. The sample size is set to $n = 10^4$ for all settings of $R$. Analogous to the synthetic data, we provide numerical results for SOLS and OLS on the running time (CPU execution time) and the optimal scaled objectives. The results are provided in Table 2. Again, we have that SOLS is much faster than OLS and they achieve comparable optimal objectives, under all settings of ranks.

Table 1: Comparison of SOLS and OLS on CPU execution time (in seconds) and the optimal scaled objective over different choices of sample sizes and noise levels on synthetic data. The results are averaged over 50 random trials, with both the mean values and standard deviations (in parentheses) provided. Note that we terminate the program after the running time exceeds $3 \times 10^4$ seconds.

| Variance of Noise | | $\sigma_z = 0$ | | | $\sigma_z = 1$ | | |
|---|---|---|---|---|---|---|---|
| Sample Size | | $n = 10^4$ | $n = 10^5$ | $n = 10^6$ | $n = 10^4$ | $n = 10^5$ | $n = 10^6$ |
| Time | OLS | 175.37 | 3683.9 | $> 3 \times 10^4$ | 168.62 | 2707.3 | $> 3 \times 10^4$ |
| | | (65.784) | (1496.7) | (NA) | (24.570) | (897.14) | (NA) |
| | SOLS | 120.34 | 128.09 | 132.93 | 121.71 | 124.84 | 128.65 |
| | | (35.711) | (37.293) | (38.649) | (34.214) | (33.774) | (32.863) |
| Objective | OLS | $< 10^{-10}$ | $< 10^{-10}$ | $< 10^{-10}$ | 0.9153 | 0.9341 | 0.9425 |
| | | ($< 10^{-10}$) | ($< 10^{-10}$) | ($< 10^{-10}$) | (0.0256) | (0.0213) | (0.0172) |
| | SOLS | $< 10^{-10}$ | $< 10^{-10}$ | $< 10^{-10}$ | 0.9376 | 0.9817 | 0.9901 |
| | | ($< 10^{-10}$) | ($< 10^{-10}$) | ($< 10^{-10}$) | (0.0261) | (0.0242) | (0.0256) |

Table 2: Comparison of SOLS and OLS on CPU execution time (in seconds) and the optimal scaled objective over different choices of ranks on the MRI data. The results are averaged over 10 random trials, with both the mean values and standard deviations (in parentheses) provided.

| | | OLS | | | SOLS | |
|---|---|---|---|---|---|---|
| Rank | $R = 3$ | $R = 5$ | $R = 10$ | $R = 3$ | $R = 5$ | $R = 10$ |
| Time | 2824.4 | 8137.2 | 26851 | 196.31 | 364.09 | 761.73 |
| | (768.08) | (1616.3) | (8320.1) | (68.180) | (145.79) | (356.76) |
| Objective | 16.003 | 11.164 | 6.8679 | 17.047 | 11.992 | 7.3968 |
| | (0.1378) | (0.1152) | (0.0471) | (0.1561) | (0.1538) | (0.0975) |

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
