[Supplementary Material · appendix.pdf]

# A  A Master Theorem

Recall that our goal is to provide sufficient conditions for the SJLT matrix $\Phi \in \mathbb{R}^{m \times n}$ to preserve the cost of all solutions for tensor regression, i.e., bounds on the sketching dimension $m$ and the per-column sparsity $s$ for which

$$\mathbb{E}_{\Phi} \sup_{x \in \mathcal{T}} \left| \|\Phi x\|_2^2 - 1 \right| < \varepsilon/10 \tag{14}$$

where $\varepsilon$ is a given precision, $\mathcal{T} = \bigcup_{E \in \mathcal{V}} \{x \in E \mid \|x\|_2 = 1\}$, and $\mathcal{V}$ is an infinite union of subspaces defined as

$$\mathcal{V} = \bigcup_{\theta_d^{(r)}, \phi_d^{(r)} \in \mathbb{R}^{p_d}, \forall r \in [R], d \in [D] \backslash \{1\}} \left\{ \operatorname{span} \left[ A^{\left\{\theta_{\backslash 1}^{(r)}\right\}}, A^{\left\{\phi_{\backslash 1}^{(r)}\right\}} \right] \right\} \text{ for the CP model, and}$$

$$\mathcal{V} = \bigcup_{\theta_d^{(r_d)}, \phi_d^{(r_d)} \in \mathbb{R}^{p_d}, \forall r_d \in [R_d], d \in [D] \backslash \{1\}} \left\{ \operatorname{span} \left[ A^{\left\{\theta_{\backslash 1}^{\{r_d\}}\right\}}, A^{\left\{\phi_{\backslash 1}^{(r_d)}\right\}} \right] \right\} \text{ for the Tucker model.}$$

Note that by linearity, it suffices to consider $x$ with $\|x\|_2 = 1$ in the above, which explains the form of (14). Also note that by Markov's inequality, (14) implies that for all $\vartheta = \operatorname{vec}(\Theta)$, where $\Theta$ follows the low-rank CP or Tucker decomposition, with probability at least 9/10, we have

$$\|\Phi A \vartheta - \Phi b\|_2^2 = (1 \pm \varepsilon)\|A\vartheta - b\|_2^2. \tag{15}$$

The next theorem follows immediately by plugging in to the bound in Section 8.5 of [2], which our work builds upon. We instantiate the conditions of that theorem for the CP model; the instantiation for the Tucker model follows analogously.

**Theorem 3.** Let $\mathcal{T} \subset \mathcal{B}_n$ and $\Phi \in \mathbb{R}^{m \times n}$ be an SJLT matrix with column sparsity $s$, and

$$p_{\mathcal{V}} = \sup_{\theta_d^{(r)} \in \mathbb{R}^{p_d}, \forall r \in [R], \ d \in [D] \backslash \{1\}} \dim \left( \operatorname{span} \left[ A^{\left\{\theta_{\backslash 1}^{(r)}\right\}}, A^{\left\{\phi_{\backslash 1}^{(r)}\right\}} \right] \right).$$

Then with probability at least 9/10, (15) holds if $m$ and $s$ satisfy

$$m \gtrsim \frac{\left( \gamma_2^2(\mathcal{V}, \rho_{\mathrm{Fin}}) + p_{\mathcal{V}} + \log \mathcal{N}(\mathcal{V}, \rho_{\mathrm{Fin}}, \varepsilon_0) \right) (\log^4 m)(\log^5 n)}{\varepsilon^2}, \tag{16}$$

$$s \gtrsim \frac{\left( \left[ \int_0^{\varepsilon_0} (\log \mathcal{N}(\mathcal{V}, \rho_{\mathrm{Fin}}, t))^{1/2} \, dt \right]^2 + \widetilde{\alpha}^2 \log^2 \mathcal{N}(\mathcal{V}, \rho_{\mathrm{Fin}}, \varepsilon_0) + \varepsilon_0^2 p_{\mathcal{V}} \log \frac{1}{\varepsilon_0} \right) (\log^6 m)(\log^5 n)}{\varepsilon^2}, \tag{17}$$

where $\widetilde{\alpha}^2$ is the largest leverage score of any $\left[ A^{\left\{\theta_{\backslash 1}^{(r)}\right\}}, A^{\left\{\phi_{\backslash 1}^{(r)}\right\}} \right] \in \mathcal{V}$ and $\mathcal{N}(\mathcal{V}, \rho_{\mathrm{Fin}}, t)$ is the covering number of $\mathcal{V}$ with radius $t$ under the Finsler metric.

*Proof.* From the main result in [2], we have that (14) holds if $m$ and $s$ satisfy

$$m \gtrsim \varepsilon^{-2}(\log^3 m)(\log^5 n)\gamma_2^2(\mathcal{V}, \rho_{\mathrm{Fin}}) + \varepsilon^{-2}(\log^4 m)(\log^5 n)\left(p_{\mathcal{V}} + \log \mathcal{N}(\mathcal{V}, \rho_{\mathrm{Fin}}, \varepsilon_0)\right),$$

$$s \gtrsim \varepsilon^{-2}(\log^4 m)(\log^5 n)\left(\widetilde{\alpha}^2 \log^2 \mathcal{N}(\mathcal{V}, \rho_{\mathrm{Fin}}, \varepsilon_0) + \varepsilon_0^2 p_{\mathcal{V}} \log \frac{1}{\varepsilon_0} + \left[\int_0^{\varepsilon_0} (\log \mathcal{N}(\mathcal{V}, \rho_{\mathrm{Fin}}, t))^{1/2} \, dt\right]^2\right)$$

$$+ \varepsilon^{-2}(\log^6 m)(\log^4 n),$$

which can be obtained from (16) and (17). Thus with probability at least 9/10, (15) holds following the argument above and we finish the proof. $\qquad\square$

# B  A Progressive Proof for Main Theorems

Given Theorem 3, the main technical difficulty lies in providing tight bounds on the various terms involved in $m$ and $s$ in Theorem 3, which depend on whether we are working in the CP model or the

Tucker model. We start with the most basic case of rank $R = 1$ for two way tensors (matrices) $D = 2$ (Theorem 4), then generalize to general ranks $R \geq 1$ for two say tensors $D = 2$ (Theorem 5), then to general tensors $D \geq 1$ with rank $R = 1$ (Theorem 6), then finally to the generic CP model with $D \geq 1$ and $R \geq 1$ (Theorem 1). This helps clarify the analysis and makes the proof of Theorem 1 straightforward. The analysis for the general Tucker model can be addressed in a similar way, and we only provide the proof for the general case to avoid redundancy.

## B.1 Base Case: Rank-1 and Two-Way Tensors

We start with the base case when $R = 1$ and $D = 2$, i.e., the parameter space is $\mathcal{S}_{2,1}$. Then the parameter admits the decomposition $\Theta = \theta_1 \circ \theta_2$. For notational convenience, we let $\Theta = u \circ v$, where $u \in \mathbb{R}^{p_1}$ and $v \in \mathbb{R}^{p_2}$, and let $A^v = \sum_{i=1}^{p_2} A^{(i)} v_i$, where $A = [A^{(1)}, \ldots, A^{(p_2)}] \in \mathbb{R}^{n \times p_2 p_1}$ with $A^{(i)} \in \mathbb{R}^{n \times p_1}$ for all $i \in [p_2]$. Consequently, the observation model (4) can be written as

$$b = A(v \otimes u) + z = A^v u + z,$$

and the corresponding OLS and SOLS using an SJLT matrix $\Phi \in \mathbb{R}^{m \times n}$ are, respectively,

$$\min_{v \in \mathbb{R}^{p_2}, u \in \mathbb{R}^{p_1}} \|A^v u - b\|_2^2 \text{ and } \min_{v \in \mathbb{R}^{p_2}, u \in \mathbb{R}^{p_1}} \|\Phi A^v u - \Phi b\|_2^2.$$

Next, we show the following theorem, which provides sufficient conditions for the base case $\mathcal{S}_{2,1}$.

**Theorem 4.** Suppose $\max_{i \in [n]} \ell_i^2(A) \leq 1/p_2^2$. Let

$$\mathcal{T} = \left\{ \frac{Ax - Ay}{\|Ax - Ay\|_2} \;\middle|\; x = v_1 \otimes u_1, y = v_2 \otimes u_2, \; u_1, u_2 \in \mathbb{R}^{p_1} \right\}$$

and $\Phi \in \mathbb{R}^{m \times n}$ be an SJLT matrix with column sparsity $s$. Then with probability at least $9/10$, (15) holds if $m$ and $s$ satisfy

$$m \gtrsim \varepsilon^{-2} (p_1 + p_2) \log \left( (p_1 + p_2) \kappa_A \right) (\log^4 m)(\log^5 n), \tag{18}$$

$$s \gtrsim \varepsilon^{-2} \log^2(p_1 + p_2)(\log^6 m)(\log^5 n). \tag{19}$$

The proof of Theorem 4 is provided in Appendix C. From Theorem 4, we have that (15) holds when $m = \Omega(p_1 + p_2)$ and $s = \Omega(1)$.

## B.2 Extension to General Ranks

We next extend our analysis to the case of two-way tensors with general rank, i.e., the parameter space is $\mathcal{S}_{2,R}$ for $R \geq 1$. In this case, we have $\Theta = \sum_{r=1}^{R} u^{(r)} \circ v^{(r)}$, where $u^{(r)} \in \mathbb{R}^{p_1}$ and $v^{(r)} \in \mathbb{R}^{p_2}$ for all $r \in [R]$, and $A^{\{v^{(r)}\}} = \left[ \sum_{i=1}^{p_2} A^{(i)} v_i^{(1)}, \ldots, \sum_{i=1}^{p_2} A^{(i)} v_i^{(R)} \right]$, where $A = [A^{(1)}, \ldots, A^{(p_2)}] \in \mathbb{R}^{n \times p_2 p_1}$ and $A^{(i)} \in \mathbb{R}^{n \times p_1}$ for all $i \in [p_2]$. Consequently, the observation model (4) can be written as

$$b = A^{\{v^{(r)}\}} \left[ u^{(1)\top} \ldots u^{(R)\top} \right]^\top + z,$$

and the corresponding OLS and SOLS using an SJLT matrix $\Phi \in \mathbb{R}^{m \times n}$ are, respectively,

$$\min_{v^{(r)} \in \mathbb{R}^{p_2}, u^{(r)} \in \mathbb{R}^{p_1}, \forall r \in [R]} \left\| A^{\{v^{(r)}\}} \left[ u^{(1)\top} \ldots u^{(R)\top} \right]^\top - b \right\|_2^2, \text{ and}$$

$$\min_{v^{(r)} \in \mathbb{R}^{p_2}, u^{(r)} \in \mathbb{R}^{p_1}, \forall r \in [R]} \left\| \Phi A^{\{v^{(r)}\}} \left[ u^{(1)\top} \ldots u^{(R)\top} \right]^\top - \Phi b \right\|_2^2.$$

Our next theorem provides sufficient conditions for $\mathcal{S}_{2,R}$.

**Theorem 5.** Suppose $R \leq p_2/2$ and $\max_{i \in [n]} \ell_i^2(A) \leq 1/(R^2 p_2^2)$. Let

$$\mathcal{T} = \left\{ \frac{Ax - Ay}{\|Ax - Ay\|_2} \;\middle|\; x = \sum_{r=1}^{R} v_1^{(r)} \otimes u_1^{(r)}, y = \sum_{r=1}^{R} v_2^{(r)} \otimes u_2^{(r)}, \; u_1^{(r)}, u_1^{(r)} \in \mathbb{R}^{p_1}, \; \forall r \in [R] \right\}$$

and $\Phi \in \mathbb{R}^{m \times n}$ be an SJLT matrix with column sparsity $s$. Then with probability at least $9/10$, (15) holds if $m$ and $s$ satisfy

$$
\begin{aligned}
m &\gtrsim \varepsilon^{-2}(\log^4 m)(\log^5 n)R\,(p_1 + p_2)\log\left(R(p_1 + p_2)\kappa_A\right), \\
s &\gtrsim \varepsilon^{-2}(\log^6 m)(\log^5 n)\log^2\left(R(p_1 + p_2)\kappa_A\right).
\end{aligned}
$$

The proof of Theorem 5 is provided in Appendix D. From Theorem 5, we have that when $m = \Omega(R(p_1 + p_2))$ and $s = \Omega(1)$, (15) holds using an SJLT matrix $\Phi$. The extra condition that $R \leq p_2/2$ is not restrictive, as in applications of low-rank tensors, typically $R \ll \min_{d \in [D]} p_d$.

## B.3   Extension to General Tensors

We first extend our analysis to general tensors with rank 1, i.e., the parameter space is now $\mathcal{S}_{D,1}$ for $D \geq 2$. In this case, we have $\Theta = \theta_1 \circ \cdots \circ \theta_D$, where $\theta_d \in \mathbb{R}^{p_d}$ for all $d \in [D]$. Consequently, the observation model (4) can be written as

$$
b = A \cdot (\theta_D \otimes \cdots \otimes \theta_1) + z = A^{\{\theta_{\backslash 1}\}} \cdot \theta_1 + z,
$$

and the corresponding OLS and SOLS using an SJLT matrix $\Phi \in \mathbb{R}^{m \times n}$ are, respectively,

$$
\min_{\theta_i \in \mathbb{R}^{p_i}, \forall i \in [D]} \left\| A^{\{\theta_{\backslash 1}\}} \theta_1 - b \right\|_2^2 \quad \text{and} \quad \min_{\theta_i \in \mathbb{R}^{p_i}, \forall i \in [D]} \left\| \Phi A^{\{\theta_{\backslash 1}\}} \theta_1 - \Phi b \right\|_2^2.
$$

Our next theorem provides sufficient conditions for $\mathcal{S}_{D,1}$.

**Theorem 6.** Suppose $\max_{i \in [n]} \ell_i^2(A) \leq 1 / \left( \sum_{d=2}^D p_d \right)^2$. For any $\vartheta = \theta_D \otimes \cdots \otimes \theta_1 \in \mathcal{S}_{\odot D,1}$ and $\varphi = \phi_D \otimes \cdots \otimes \phi_1 \in \mathcal{S}_{\odot D,1}$, $\theta_d, \phi_d \in \mathbb{R}^{p_d}$ for all $d \in [D]$. Let

$$
\mathcal{T} = \left\{ \frac{A\vartheta - A\varphi}{\|A\vartheta - A\varphi\|_2} \,\middle|\, \vartheta = \theta_D \otimes \cdots \otimes \theta_1, \varphi = \phi_D \otimes \cdots \otimes \phi_1, \, \theta_d, \phi_d \in \mathbb{R}^{p_d}, \, \forall d \in [D] \right\}
$$

and $\Phi \in \mathbb{R}^{m \times n}$ be an SJLT matrix with column sparsity $s$. Then with probability at least $9/10$, (15) holds if $m$ and $s$ satisfy

$$
\begin{aligned}
m &\gtrsim \varepsilon^{-2}(\log^4 m)(\log^5 n)\left( \sum_{d=1}^D p_d \log\left( D\kappa_A \sum_{d=1}^D p_d \right) \right), \\
s &\gtrsim \varepsilon^{-2}(\log^6 m)(\log^5 n)\log^2\left( \sum_{d=1}^D p_d \right).
\end{aligned}
$$

The proof of Theorem 6 is provided in Appendix E. From Theorem 6, we have that when $m = \Omega\left( \sum_{d=1}^D p_d \right)$ and $s = \Omega(1)$, (15) holds using an SJLT matrix $\Phi$.

# C   Proof of Theorem 4

We start with an illustration that the set $\mathcal{T}$ can be reparameterized to the following set with respect to tensors with orthogonal factors:

$$
\mathcal{T} = \bigcup_{E \in \mathcal{V}} \{x \in E \mid \|x\|_2 = 1\}, \quad \text{where } \mathcal{V} = \bigcup_{\widetilde{\mathcal{W}}} \{\operatorname{span}[A^{v_1}, A^{v_2}]\} \text{ and}
$$

$$
\widetilde{\mathcal{W}} = \{v_1, v_2 \in \mathcal{B}_{p_2}, \langle v_1, v_2 \rangle = 0\}.
$$

Suppose $\langle v_1, v_2 \rangle \neq 0$. Let $v_2 = \alpha v_1 + \beta z$ for some $\alpha, \beta \in \mathbb{R}$ and a unit vector $z \in \mathbb{R}^{p_2}$, where $\langle v_1, z \rangle = 0$. Then we have

$$
\frac{Ax - Ay}{\|Ax - Ay\|_2} = \frac{A^{v_1}u_1 - A^{v_2}u_2}{\|A^{v_1}u_1 - A^{v_2}u_2\|_2} = \frac{A^{v_1}u_1 - A^{\alpha v_1 + \beta z}u_2}{\|A^{v_1}u_1 - A^{\alpha v_1 + \beta z}u_2\|_2} = \frac{A^{v_1}u_1 - A^{\alpha v_1}u_2 - A^{\beta z}u_2}{\|A^{v_1}u_1 - A^{\alpha v_1}u_2 - A^{\beta z}u_2\|_2}
$$

$$
= \frac{A^{v_1}(u_1 - \alpha u_2) - A^z(\beta u_2)}{\|A^{v_1}(u_1 - \alpha u_2) - A^z(\beta u_2)\|_2},
$$

which is equivalent to $\langle v_1, v_2 \rangle = 0$ by reparameterizing $z$ as $v_2$.

Next, by Theorem 3, we need to upper bound $\rho_{\mathcal{V}}$, $\gamma_2^2(\mathcal{V}, \rho_{\text{Fin}})$, and $\mathcal{N}(\mathcal{V}, \rho_{\text{Fin}}, \varepsilon_0)$. These will be addressed separately as follows.

**Part 1: Bound $p_{\mathcal{V}}$.** For notational convenience, we denote $A^{v_1, v_2} = [A^{v_1}, A^{v_2}]$. It is straightforward that

$$
p_{\mathcal{V}} = \sup_{v_1, v_2 \in \mathcal{B}_{p_2}, \langle v_1, v_2 \rangle = 0} \dim \{\text{span}(A^{v_1, v_2})\} \leq 2p_1. \tag{20}
$$

**Part 2: Bound $\gamma_2^2(\mathcal{V}, \rho_{\text{Fin}})$.** By the definition of the $\gamma_2$-functional in (7) for the Finsler metric, we have

$$
\gamma_2(\mathcal{V}, \rho_{\text{Fin}}) = \inf_{\{\overline{\mathcal{V}}_k\}_{k=0}^{\infty}} \sup_{A^{v_1, v_2} \in \mathcal{V}} \sum_{k=0}^{\infty} 2^{k/2} \cdot \rho_{\text{Fin}}(A^{v_1, v_2}, \overline{\mathcal{V}}_k),
$$

where $\overline{\mathcal{V}}_k$ is an $\varepsilon_k$-net of $\mathcal{V}$, i,e., for any $A^{v_1, v_2} \in \mathcal{V}$ there exist $\overline{v}_1, \overline{v}_2 \in \mathcal{B}_{p_2}$ with $\langle \overline{v}_1, \overline{v}_2 \rangle = 0$, $\|v_1 - \overline{v}_1\|_2 \leq \eta_k$, and $\|v_2 - \overline{v}_2\|_2 \leq \eta_k$, such that $A^{\overline{v}_1, \overline{v}_2} \in \overline{\mathcal{V}}_k$ and $\rho_{\text{Fin}}(A^{v_1, v_2}, A^{\overline{v}_1, \overline{v}_2}) \leq \varepsilon_k$.

From Lemma 6, we have $\rho_{\text{Fin}}(A^{v_1, v_2}, \overline{\mathcal{V}}_k) \leq 2\kappa_A \eta_k$ for $\|v_1 - \overline{v}_1\|_2 \leq \eta_k$ and $\|v_2 - \overline{v}_2\|_2 \leq \eta_k$. On the other hand, we have that $\rho_{\text{Fin}}(A^{v_1, v_2}, \overline{\mathcal{V}}_k) \leq 1$ always holds. Therefore, we have

$$
\rho_{\text{Fin}}(A^{v_1, v_2}, \overline{\mathcal{V}}_k) \leq \min\{2\kappa_A \eta_k, 1\}.
$$

Let $k'$ be the smallest integer such that $2\kappa_A \eta_{k'} \leq 1$. Then we have

$$
\gamma_2(\mathcal{V}, \rho_{\text{Fin}}) \leq \sum_{k=0}^{\infty} 2^{k/2} \cdot \rho_{\text{Fin}}(A^{v_1, v_2}, \overline{\mathcal{V}}_k) \leq \sum_{k=0}^{k'} 2^{k/2} + \sum_{k=k'+1}^{\infty} 2^{k/2} \cdot \rho_{\text{Fin}}(A^{v_1, v_2}, \overline{\mathcal{V}}_k). \tag{21}
$$

Suppose that $\eta_0 = 1$. Then we have $|\overline{\mathcal{V}}_0| = 1$. For $k \geq 1$, we have $\eta_k < 1$ and $|\overline{\mathcal{V}}_k| \leq (3/\eta_k)^{p_2}$ [29]. By the definition of admissible sequences in the $\gamma_2$-functional, we require $|\overline{\mathcal{V}}_k| \leq 2^{2^k}$. Without loss of generality, suppose that for all $k \leq k'$, we have $|\overline{\mathcal{V}}_k| \leq 2^{2^k} \leq (3/\eta_k)^{p_2}$. Then we have $2^{k/2} \leq \sqrt{p_2 \log \frac{3}{\eta_k}}$, which implies

$$
\sum_{k=0}^{k'} 2^{k/2} = \frac{2^{k'/2}}{\sqrt{2} - 1} \lesssim \sqrt{p_2 \log \frac{1}{\eta_{k'}}}. \tag{22}
$$

For $k > k'$, suppose we choose $\eta_{k+1} = \eta_k^2$. Then we have

$$
\left(\frac{3}{\eta_{k+1}}\right)^{p_2} \leq \left(\frac{3}{\eta_k}\right)^{2p_2} \leq \left(2^{2^k}\right)^2 = 2^{2^{k+1}}, \tag{23}
$$

which implies $|\overline{\mathcal{V}}_{k+1}| \leq 2^{2^{k+1}}$ as long as $|\overline{\mathcal{V}}_{k+1}| \leq (3/\eta_{k+1})^{p_2}$ holds. In other words, we have $|\overline{\mathcal{V}}_k| \leq 2^{2^k}$ if we choose $\eta_{k+1} = \eta_k^2$ for all $k > k'$. Suppose $k'$ is the smallest integer such that when we choose $\eta_{k'+1} = \frac{1}{4\kappa_A}$, then $\left(\frac{3}{\eta_{k'+1}}\right)^{p_2} \leq 2^{2^{k'+1}}$ holds. This implies (23) holds and

$\rho_{\text{Fin}}(A^{v_1,v_2}, \overline{\mathcal{V}}_k) \leq (1/2)2^{k-k'}$ for all $k > k'$. Then we have

$$\sum_{k=k'+1}^{\infty} 2^{k/2} \cdot \rho_{\text{Fin}}(A^{v_1,v_2}, \overline{\mathcal{V}}_k) = 2^{k'/2} \cdot \sum_{t=1}^{\infty} 2^{t/2} \cdot \left(\frac{1}{2}\right)^{2^t} \leq 2^{k'/2} \lesssim \sqrt{p_2 \log \frac{1}{\eta_{k'}}}, \qquad (24)$$

where the first inequality is from the Cauchy condensation test $\sum_{t=0}^{\infty} 2^{t/2} \cdot \left(\frac{1}{2}\right)^{2^t} \leq 2 \cdot \sum_{t=0}^{\infty} \left(\frac{1}{2}\right)^t = 1$ and the second inequality is from (22).

Combining (21), (22), and (24), we have

$$\gamma_2^2(\mathcal{V}, \rho_{\text{Fin}}) \lesssim p_2 \log \frac{1}{\eta_{k'}}. \qquad (25)$$

From Lemma 6, suppose we choose a small enough $\varepsilon_0$ such that $\varepsilon_0 \leq 2\kappa_A \eta_{k'}$. Then (25) implies

$$\gamma_2^2(\mathcal{V}, \rho_{\text{Fin}}) \lesssim p_2 \log \frac{\kappa_A}{\varepsilon_0}. \qquad (26)$$

**Part 3: Bound** $\mathcal{N}(\mathcal{V}, \rho_{\text{Fin}}, \varepsilon_0)$. From our choice from Part 2, $\varepsilon_0 \in (0,1)$ is a constant. Then it is straightforward that

$$\mathcal{N}(\mathcal{V}, \rho_{\text{Fin}}, \varepsilon_0) \leq \left(\frac{3}{\varepsilon_0}\right)^{2p_2}. \qquad (27)$$

This implies

$$\int_0^{\varepsilon_0} [\log \mathcal{N}(\mathcal{V}, \rho_{\text{Fin}}, t)]^{1/2} dt \leq \int_0^{\varepsilon_0} (\log (3/t)^{p_2})^{1/2} dt \overset{(i)}{\lesssim} \sqrt{p_2} \int_0^{\varepsilon_0} (-\log t)^{1/2} dt$$

$$= \sqrt{p_2} \int_{-\infty}^{(-\log \varepsilon_0)^{1/2}} 2w^2 e^{-w^2} dw = \sqrt{p_2} \left( \left[w \cdot e^{-w^2}\right]_{-\infty}^{(-\log \varepsilon_0)^{1/2}} - \int_{-\infty}^{(-\log \varepsilon_0)^{1/2}} e^{-w^2} dw \right)$$

$$\leq \sqrt{p_2} \left[w \cdot e^{-w^2}\right]_{-\infty}^{(-\log \varepsilon_0)^{1/2}} = \varepsilon_0 \sqrt{p_2 \log \frac{1}{\varepsilon_0}}. \qquad (28)$$

where $(i)$ is from setting $w = (-\log t)^{1/2}$. From Lemma 4, we have

$$\widetilde{\alpha}^2 = \max_{i \in [n]} \ell_i^2(A^{v_1,v_2}) \leq \max_{i \in [n]} \ell_i^2(A) \leq 1/p_2^2. \qquad (29)$$

Combining (20), (26)–(29), and Theorem 3, we have that the claim holds if

$$m \gtrsim \varepsilon^{-2} \left(p_2 \log \frac{\kappa_A}{\varepsilon_0} + p_1 + p_2 \log \frac{1}{\varepsilon_0}\right)(\log^4 m)(\log^5 n) \quad \text{and}$$

$$s \gtrsim \varepsilon^{-2} \left(\log^2 \frac{1}{\varepsilon_0} + \varepsilon_0^2(p_1 + p_2) \log \frac{1}{\varepsilon_0}\right)(\log^6 m)(\log^5 n).$$

Taking $\varepsilon_0 = 1/(p_1 + p_2)$, we finish the proof. Note that since $2\kappa_A \eta_{k'} \geq 1/2$, we only require $\rho_{\text{Fin}}(A^{v_1,v_2}, \overline{\mathcal{V}}_{k'}) \leq 1/2$ in Part 2. Thus the choice $\varepsilon_0 = 1/(p_1 + p_2)$ is valid here.

## D  Proof of Theorem 5

Denote $A^{\{v_i^{(r)}\}} = \left[A^{\{v_1^{(r)}\}}, A^{\{v_2^{(r)}\}}\right] \in \mathbb{R}^{n \times 2Rp_1}$. We illustrate that the set $\mathcal{T}$ can be reparameterized to the following set with respect to tensors with partial orthogonal factors:

$$\mathcal{T} = \bigcup_{E \in \mathcal{V}} \{x \in E \mid \|x\|_2 = 1\}, \quad \text{where } \mathcal{V} = \bigcup_{\widetilde{\mathcal{W}}} \text{span}\left(A^{\{v_i^{(r)}\}}\right) \quad \text{and}$$

$$\widetilde{\mathcal{W}} = \left\{\forall i \in [2],\ r, q \in [R],\ q \neq r, v_i^{(r)} \in \mathcal{B}_{p_2}, \langle v_1^{(r)}, v_2^{(r)}\rangle = \langle v_i^{(r)}, v_i^{(q)}\rangle = 0\right\}.$$

Suppose for all $r \in [R]$, $v_2^{(r)} = \alpha^{(r)} v_1^{(r)} + \beta^{(r)} z^{(r)}$ for some $\alpha^{(r)}, \beta^{(r)} \in \mathbb{R}$ and unit vectors $z^{(r)} \in \mathbb{R}^{p_2}$, where $\langle v_1^{(r)}, z^{(r)} \rangle = 0$. Then we have

$$
\begin{aligned}
Ax - Ay &= \sum_{r=1}^{R} \left( A^{v_1^{(r)}} \cdot u_1^{(r)} - A^{v_2^{(r)}} \cdot u_2^{(r)} \right) = \sum_{r=1}^{R} \left( A^{v_1^{(r)}} \cdot u_1^{(r)} - A^{\alpha^{(r)} v_1^{(r)} + \beta^{(r)} z^{(r)}} \cdot u_2^{(r)} \right) \\
&= \sum_{r=1}^{R} \left( A^{v_1^{(r)}} \cdot u_1^{(r)} - A^{\alpha^{(r)} v_1^{(r)}} \cdot u_2^{(r)} - A^{\beta^{(r)} z^{(r)}} \cdot u_2^{(r)} \right) \\
&= \sum_{r=1}^{R} \left( A^{v_1^{(r)}} \cdot \left( u_1^{(r)} - \alpha^{(r)} u_2^{(r)} \right) - A^{z^{(r)}} \cdot \left( \beta^{(r)} u_2^{(r)} \right) \right).
\end{aligned}
$$

which is equivalent to $\langle v_1^{(r)}, v_2^{(r)} \rangle = 0$ by reparameterizing $z^{(r)}$ as $v_2^{(r)}$.

Using a similar argument, we show the general scenario. For any $r \in [R]$, $r \geq 2$, w.l.o.g., suppose

$$
v_1^{(r)} = \alpha_1^{(r,1)} v_1^{(1)} + \sum_{i=2}^{r} \alpha_1^{(r,i)} z_1^{(i)} \quad \text{and} \quad v_2^{(r)} = \beta_1^{(r,1)} v_1^{(1)} + \sum_{i=2}^{r} \beta_1^{(r,i)} z_1^{(i)} + \sum_{j=1}^{r} \beta_2^{(r,j)} z_2^{(j)}.
$$

where $\alpha_1^{(r,i)}, \beta_1^{(r,i)}, \beta_2^{(r,j)} \in \mathbb{R}$ are real coefficients and $\langle v_1^{(1)}, z_1^{(i)} \rangle = \langle v_1^{(1)}, z_2^{(i)} \rangle = \langle z_1^{(i)}, z_2^{(j)} \rangle = 0$ for any $i, j \in [r]$. For $R = 1$, the argument is identical to the one above. For $2 \leq R \leq p_2/2$, we have

$$
\begin{aligned}
Ax - Ay &= \sum_{r=1}^{R} \left( A^{v_1^{(r)}} \cdot u_1^{(r)} - A^{v_2^{(r)}} \cdot u_2^{(r)} \right) \\
&= \sum_{r=2}^{R} \left( A^{\alpha_1^{(r,1)} v_1^{(1)} + \sum_{i=2}^{r} \alpha_1^{(r,i)} z_1^{(i)}} \cdot u_1^{(r)} - A^{\beta_1^{(r,1)} v_1^{(1)} + \sum_{i=2}^{r} \beta_1^{(r,i)} z_1^{(i)} + \sum_{j=1}^{r} \beta_2^{(r,j)} z_2^{(j)}} \cdot u_2^{(r)} \right) \\
&\qquad\qquad\qquad\qquad + A^{v_1^{(1)}} \cdot u_1^{(r)} - A^{\left( \beta_1^{(1,1)} v_1^{(1)} + \beta_2^{(1,1)} z_2^{(1)} \right)} \cdot u_2^{(r)} \\
&= \sum_{r=2}^{R} \left( A^{v_1^{(1)}} \cdot \left( \alpha_1^{(r,1)} u_1^{(1)} - \beta_1^{(r,1)} u_2^{(1)} \right) + \sum_{i=2}^{r} A^{z_1^{(i)}} \cdot \left( \alpha_1^{(r,i)} u_1^{(i)} - \beta_1^{(r,i)} u_2^{(i)} \right) \right. \\
&\qquad\qquad \left. - \sum_{j=1}^{r} A^{z_2^{(j)}} \cdot \left( \beta_2^{(r,j)} u_2^{(j)} \right) \right) + A^{v_1^{(1)}} \cdot u_1^{(r)} - A^{\left( \beta_1^{(1,1)} v_1^{(1)} + \beta_2^{(1,1)} z_2^{(1)} \right)} \cdot u_2^{(r)},
\end{aligned}
$$

which is equivalent to $\langle v_i^{(r)}, v_i^{(q)} \rangle = 0$ and $\langle v_1^{(r)}, v_2^{(r)} \rangle = 0$ for all $i \in [2]$, $r \in [R]$, and $q \neq r$ by reparameterizing $z_1^{(i)}$ as $v_1^{(i)}$ and $z_2^{(j)}$ as $v_2^{(j)}$.

Next, analogous to Theorem 4, we analyze upper bounds on $\rho_{\mathcal{V}}$, $\gamma_2^2(\mathcal{V}, \rho_{\text{Fin}})$, and $\mathcal{N}(\mathcal{V}, \rho_{\text{Fin}}, \varepsilon_0)$, and obtain the result from Theorem 3.

**Part 1: Bound $p_{\mathcal{V}}$.** It is straightforward that

$$
p_{\mathcal{V}} = \sup_{\widetilde{\mathcal{W}}} \dim \left\{ \text{span} \left( A^{\left\{ v_i^{(r)} \right\}} \right) \right\} \leq 2Rp_1. \tag{30}
$$

**Part 2: Bound $\gamma_2^2(\mathcal{V}, \rho_{\text{Fin}})$.** The $\gamma_2$-functional in this case is

$$
\gamma_2^2(\mathcal{V}, \rho_{\text{Fin}}) = \inf_{\{\overline{\mathcal{V}}_k\}_{k=0}^{\infty}} \sup_{A^{\left\{ v_i^{(r)} \right\}} \in \mathcal{V}} \sum_{k=0}^{\infty} 2^{r/2} \cdot \rho_{\text{Fin}} \left( A^{\left\{ v_i^{(r)} \right\}}, \overline{\mathcal{V}}_k \right),
$$

where $\overline{\mathcal{V}}_k$ is an $\varepsilon_k$-net of $\mathcal{V}$.

Following the same argument in Part 2 of the proof for Theorem 4, we have from Lemma 7 that if $k'$ is the smallest integer such that $2R\kappa_A \eta_{k'} \leq 1$ and we choose $\eta_{k'+1} = \frac{1}{4R\kappa_A}$, then we choose a small

enough $\varepsilon_0$ such that $\varepsilon_0 \le 2R\kappa_A \eta_{k'}$,

$$\gamma_2^2(\mathcal{V}, \rho_{\text{Fin}}) \lesssim Rp_2 \log \frac{R\kappa_A}{\varepsilon_0}. \tag{31}$$

**Part 3: Bound** $\mathcal{N}(\mathcal{V}, \rho_{\textbf{Fin}}, \varepsilon_0)$. It is straightforward that $\mathcal{N}(\mathcal{V}, \rho_{\text{Fin}}, \varepsilon_0) \le \left(\frac{3}{\varepsilon_0}\right)^{2Rp_2}$. Following the same argument in Part 3 of the proof for Theorem 4, we have

$$\int_0^{\varepsilon_0} [\log \mathcal{N}(\mathcal{V}, \rho_{\text{Fin}}, t)]^{1/2} dt \lesssim \varepsilon_0 \sqrt{Rp_2 \log \frac{1}{\varepsilon_0}}. \tag{32}$$

From Lemma 5, we have

$$\widetilde{\alpha}^2 = \max_{i \in [n]} \ell_i^2 \left( A^{\left\{v_i^{(r)}\right\}} \right) \le \max_{i \in [n]} \ell_i^2(A) \le 1/(R^2 p_2^2). \tag{33}$$

Combining (30) – (33) and Theorem 3, we have that the claim holds if

$$m \gtrsim \varepsilon^{-2} R \left( p_2 \log \frac{R\kappa_A}{\varepsilon_0} + p_1 + p_2 \log \frac{1}{\varepsilon_0} \right) (\log^4 m)(\log^5 n) \text{ and}$$

$$s \gtrsim \varepsilon^{-2} \left( \log^2 \frac{1}{\varepsilon_0} + \varepsilon_0^2 R(p_1 + p_2) \log \frac{1}{\varepsilon_0} \right) (\log^6 m)(\log^5 n).$$

We finish the proof by taking $\varepsilon_0 = \frac{1}{R(p_1+p_2)}$. Note that this choice of $\varepsilon$ satisfies the requirement in Part 2.

# E  Proof of Theorem 6

Denote $\vartheta_{\backslash 1} = \theta_D \otimes \cdots \otimes \theta_2$, $\varphi_{\backslash 1} = \phi_D \otimes \cdots \otimes \phi_2$ and $A^{\vartheta_{\backslash 1}, \varphi_{\backslash 1}} = \left[A^{\left\{\theta_{\backslash 1}\right\}}, A^{\left\{\phi_{\backslash 1}\right\}}\right] \in \mathbb{R}^{n \times 2p_1}$. We illustrate that the set $\mathcal{T}$ can be reparameterized to the following set with respect to tensors with partial orthogonal factors:

$$\mathcal{T} = \bigcup_{E \in \mathcal{V}} \{x \in E \mid \|x\|_2 = 1\}, \quad \text{where } \mathcal{V} = \bigcup_{\widetilde{\mathcal{W}}} \text{span} \left(A^{\vartheta_{\backslash 1}, \varphi_{\backslash 1}}\right) \text{ and}$$

$$\widetilde{\mathcal{W}} = \{\forall d \in [D]\backslash\{1\}, \ \theta_d, \phi_d \in \mathcal{B}_{p_d}, \exists i \in [D]\backslash\{1\} \text{ s.t. } \langle \theta_i, \phi_i \rangle = 0\},$$

W.l.o.g., suppose $\phi_D = \alpha\theta_D + \beta z$ for some $\alpha, \beta \in \mathbb{R}$ and a unit vector $z \in \mathbb{R}^{p_D}$, where $\langle \theta_D, z \rangle = 0$. Then we have

$$
\begin{aligned}
A\vartheta - A\varphi &= A^{\left\{\theta_{\backslash 1}\right\}}\theta_1 - A^{\left\{\phi_{\backslash 1}\right\}}\phi_1 = A(\theta_D \otimes \cdots \otimes \theta_2 \otimes I_{p_1})\theta_1 - A(\phi_D \otimes \cdots \otimes \phi_2 \otimes I_{p_1})\phi_1 \\
&= A(\theta_D \otimes \cdots \otimes \theta_2 \otimes I_{p_1})\theta_1 - A((\alpha\theta_D + \beta z) \otimes \phi_{D-1} \otimes \cdots \otimes \phi_2 \otimes I_{p_1})\phi_1 \\
&= A(\theta_D \otimes \cdots \otimes \theta_2 \otimes I_{p_1})\theta_1 - A(\alpha\theta_D \otimes \cdots \otimes \phi_2 \otimes I_{p_1})\phi_1 - A(\beta z \otimes \cdots \otimes \phi_2 \otimes I_{p_1})\phi_1 \\
&= A^{\theta_D} \left(\theta_{D-1} \otimes \cdots \otimes \theta_1 - \alpha\phi_{D-1} \otimes \cdots \otimes \phi_1\right) - A^z \left(\phi_{D-1} \otimes \cdots \otimes \phi_1\right),
\end{aligned}
$$

This is equivalent to $\langle \theta_D, \phi_D \rangle = 0$ by reparameterizing $z$ as $\phi_D$.

Next, analogous to Theorem 4, we analyze upper bounds on $\rho_{\mathcal{V}}$, $\gamma_2^2(\mathcal{V}, \rho_{\text{Fin}})$, and $\mathcal{N}(\mathcal{V}, \rho_{\text{Fin}}, \varepsilon_0)$, and obtain the result from Theorem 3.

**Part 1: Bound** $p_{\mathcal{V}}$. It is straightforward that

$$p_{\mathcal{V}} = \sup_{\widetilde{\mathcal{W}}} \dim \left\{\text{span} \left(A^{\vartheta_{\backslash 1}, \varphi_{\backslash 1}}\right)\right\} \le 2p_1. \tag{34}$$

**Part 2: Bound** $\gamma_2^2(\mathcal{V}, \rho_{\textbf{Fin}})$. The $\gamma_2$-functional in this case is

$$\gamma_2^2(\mathcal{V}, \rho_{\text{Fin}}) = \inf_{\{\overline{\mathcal{V}}_k\}_{k=0}^{\infty}} \sup_{A^{\vartheta_{\backslash 1}, \varphi_{\backslash 1}} \in \mathcal{V}} \sum_{k=0}^{\infty} 2^{r/2} \cdot \rho_{\text{Fin}} \left(A^{\vartheta_{\backslash 1}, \varphi_{\backslash 1}}, \overline{\mathcal{V}}_k\right),$$

where $\overline{\mathcal{V}}_k$ is an $\varepsilon_k$-net of $\mathcal{V}$.

Following the same argument in Part 2 of the proof of Theorem 4, we have from Lemma 8 that if $k'$ is the smallest integer such that $2\kappa_A \left( (1 + \eta_{k'})^D - 1 \right) \leq 1$, then we choose $\varepsilon_0$ small enough such that

$$\varepsilon_0 \leq 2\kappa_A D \eta_{k'} \leq 2\kappa_A \left( (1 + \eta_{k'})^D - 1 \right).$$

where the second inequality is from the binomial expansion. Then we have

$$\gamma_2^2(\mathcal{V}, \rho_{\text{Fin}}) \lesssim \sum_{d=2}^{D} p_d \cdot \log \frac{D\kappa_A}{\varepsilon_0}. \tag{35}$$

**Part 3: Bound** $\mathcal{N}(\mathcal{V}, \rho_{\text{Fin}}, \varepsilon_0)$. It is straightforward that $\mathcal{N}(\mathcal{V}, \rho_{\text{Fin}}, \varepsilon_0) \leq \left( \frac{3}{\varepsilon_0} \right)^{2 \sum_{d=2}^{D} p_d}$. Following the same argument in Part 3 of the proof for Theorem 4, we have

$$\int_0^{\varepsilon_0} [\log \mathcal{N}(\mathcal{V}, \rho_{\text{Fin}}, t)]^{1/2} dt \lesssim \varepsilon_0 \sqrt{\sum_{d=2}^{D} p_d \log \frac{1}{\varepsilon_0}}. \tag{36}$$

From Lemma 5, we have

$$\widetilde{\alpha}^2 = \max_{i \in [n]} \ell_i^2 \left( A^{\vartheta_{\backslash 1}, \varphi_{\backslash 1}} \right) \leq \max_{i \in [n]} \ell_i^2(A) \leq \frac{1}{\left( \sum_{d=2}^{D} p_d \right)^2}. \tag{37}$$

Combining (34) – (37) and Theorem 3, we have that the claim holds if

$$m \gtrsim \varepsilon^{-2} \left( p_1 + \sum_{d=2}^{D} p_d \cdot \log \frac{D\kappa_A}{\varepsilon_0} \right) (\log^4 m)(\log^5 n) \text{ and}$$

$$s \gtrsim \varepsilon^{-2} \left( \log^2 \frac{1}{\varepsilon_0} + \varepsilon_0^2 \sum_{d=1}^{D} p_d \log \frac{1}{\varepsilon_0} \right) (\log^6 m)(\log^5 n).$$

We finish the proof by taking $\varepsilon_0 = \frac{1}{\sum_{d=1}^{D} p_d}$. Note that this choice of $\varepsilon$ satisfies the requirement in Part 2.

# F   Proof of Theorem 1

Denote $A^{\left\{ \vartheta_{\backslash 1}^{(r)}, \varphi_{\backslash 1}^{(r)} \right\}} = \left[ A^{\left\{ \theta_{\backslash 1}^{(r)} \right\}}, A^{\left\{ \phi_{\backslash 1}^{(r)} \right\}} \right]$. We illustrate that the set $\mathcal{T}$ can be reparameterized to the following set with respect to tensors with partial orthogonal factors:

$$\mathcal{T} = \bigcup_{E \in \mathcal{V}} \{ x \in E \mid \|x\|_2 = 1 \}, \text{ where } \mathcal{V} = \bigcup_{\widehat{\mathcal{W}}} \text{span} \left( A^{\left\{ \vartheta_{\backslash 1}^{(r)}, \varphi_{\backslash 1}^{(r)} \right\}} \right),$$

$$\widetilde{\mathcal{W}} = \Big\{ \forall r \in [R], d \in [D]\backslash\{1\}, \theta_d^{(r)}, \phi_d^{(r)} \in \mathcal{B}_{p_d}; \forall r, q \in [R], \exists i \in [D]\backslash\{1\} \text{ s.t. } \langle \theta_i^{(r)}, \phi_i^{(q)} \rangle = 0;$$

$$\forall r \in [R-1], q \in [R]\backslash[r], \exists j, k \in [D]\backslash\{1\} \text{ s.t. } \langle \theta_j^{(r)}, \theta_j^{(q)} \rangle = \langle \phi_k^{(r)}, \phi_k^{(q)} \rangle = 0 \Big\}.$$

For $R = 1$, the argument is identical to the analysis in Theorem 6. For any $r \in [R]$, $r \geq 2$, w.l.o.g., suppose

$$\theta_D^{(r)} = \alpha_1^{(r,1)} \theta_D^{(1)} + \sum_{i=2}^{r} \alpha_1^{(r,i)} z_1^{(i)} \text{ and } \phi_D^{(r)} = \beta_1^{(r,1)} \theta_D^{(1)} + \sum_{i=2}^{r} \beta_1^{(r,i)} z_1^{(i)} + \sum_{j=1}^{r} \beta_2^{(r,j)} z_2^{(j)},$$

where $\alpha_1^{(r,i)}, \beta_1^{(r,i)}, \beta_2^{(r,j)} \in \mathbb{R}$ are real coefficients and $\langle \theta_D^{(1)}, z_1^{(i)} \rangle = \langle \theta_D^{(1)}, z_2^{(i)} \rangle = \langle z_1^{(i)}, z_2^{(j)} \rangle = 0$ for any $i, j \in [r]$. Then for $2 \le R \le p_2/2$, we have

$$
A\vartheta - A\varphi = A \cdot \sum_{r=1}^{R} \left( \theta_D^{(r)} \otimes \cdots \otimes \theta_2^{(r)} \otimes I_{p_1} \right) \theta_1^{(r)} - A \cdot \sum_{r=1}^{R} \left( \phi_D^{(r)} \otimes \cdots \otimes \phi_2^{(r)} \otimes I_{p_1} \right) \phi_1^{(r)}
$$

$$
= A \cdot \sum_{r=2}^{R} \left( \left( \alpha_1^{(r,1)} \theta_D^{(1)} + \sum_{i=2}^{r} \alpha_1^{(r,i)} z_1^{(i)} \right) \otimes \cdots \otimes \theta_1^{(r)} \right) + A \cdot \left( \theta_D^{(1)} \otimes \cdots \otimes \theta_1^{(1)} \right)
$$

$$
- A \cdot \sum_{r=2}^{R} \left( \left( \beta_1^{(r,1)} \theta_D^{(1)} + \sum_{i=2}^{r} \beta_1^{(r,i)} z_1^{(i)} + \sum_{j=1}^{r} \beta_2^{(r,j)} z_2^{(j)} \right) \otimes \cdots \otimes \phi_1^{(r)} \right)
$$

$$
- A \cdot \left( \left( \beta_1^{(1,1)} \theta_D^{(1)} + \beta_2^{(1,1)} z_2^{(1)} \right) \otimes \cdots \otimes \phi_1^{(1)} \right)
$$

$$
= \sum_{r=r}^{R} A^{\theta_D^{(1)}} \left( \alpha_1^{(r,1)} \theta_{D-1}^{(r)} \otimes \cdots \otimes \theta_1^{(r)} - \beta_1^{(r,1)} \phi_{D-1}^{(r)} \otimes \cdots \otimes \phi_1^{(r)} \right)
$$

$$
+ \sum_{r=2}^{R} \sum_{i=2}^{r} A^{z_1^{(1)}} \left( \alpha_1^{(r,i)} \theta_{D-1}^{(r)} \otimes \cdots \otimes \theta_1^{(r)} - \beta_1^{(r,i)} \phi_{D-1}^{(r)} \otimes \cdots \otimes \phi_1^{(r)} \right)
$$

$$
- \sum_{r=1}^{R} \sum_{j=1}^{r} A^{z_2^{(j)}} \left( \beta_2^{(r,j)} \phi_{D-1}^{(r)} \otimes \cdots \otimes \phi_1^{(r)} \right)
$$

where $\alpha_1^{(\cdot,1)} = 1$. This is equivalent to $\langle \theta_D^{(r)}, \phi_D^{(r)} \rangle = 0$, $\langle \theta_D^{(r)}, \theta_D^{(q)} \rangle = 0$, and $\langle \phi_D^{(r)}, \phi_D^{(q)} \rangle = 0$ for all $r \in [R]$ and $q \ne [R] \backslash [r]$, by reparameterizing $z_1^{(i)}$ and $z_2^{(j)}$ as $\theta_D^{(i)}$ and $\phi_D^{(j)}$ properly. The remaining pairs of orthogonality in $\widetilde{\mathcal{W}}$ can be checked analogously by repeating the argument above.

**Part 1: Bound $p_{\mathcal{V}}$.** It is straightforward that

$$
p_{\mathcal{V}} = \sup_{\widetilde{\mathcal{W}}} \dim \left\{ \text{span} \left( A^{\left\{ \vartheta_{\backslash 1}^{(r)}, \varphi_{\backslash 1}^{(r)} \right\}} \right) \right\} \le 2R p_1. \tag{38}
$$

**Part 2: Bound $\gamma_2^2(\mathcal{V}, \rho_{\mathbf{Fin}})$.** The $\gamma_2$-functional in this case is

$$
\gamma_2^2(\mathcal{V}, \rho_{\mathbf{Fin}}) = \inf_{\{\overline{\mathcal{V}}_k\}_{k=0}^{\infty}} \sup_{A^{\left\{ \vartheta_{\backslash 1}^{(r)}, \varphi_{\backslash 1}^{(r)} \right\}} \in \mathcal{V}} \sum_{k=0}^{\infty} 2^{r/2} \cdot \rho_{\mathbf{Fin}} \left( A^{\left\{ \vartheta_{\backslash 1}^{(r)}, \varphi_{\backslash 1}^{(r)} \right\}}, \overline{\mathcal{V}}_k \right),
$$

where $\overline{\mathcal{V}}_k$ is an $\varepsilon_k$-net of $\mathcal{V}$.

Following the same argument in Part 2 of the proof for Theorem 4, we have from Lemma 9 that if $k'$ is the smallest integer such that $2R\kappa_A \left( (1 + \eta_{k'})^D - 1 \right) \le 1$, then we choose $\varepsilon_0$ small enough such that

$$
\varepsilon_0 \le 2RD\kappa_A \eta_{k'} \le 2R\kappa_A \left( (1 + \eta_{k'})^D - 1 \right),
$$

where the second inequality follows from the binomial expansion. Then we have

$$
\gamma_2^2(\mathcal{V}, \rho_{\mathbf{Fin}}) \lesssim \sum_{d=2}^{D} p_d \cdot \log \frac{RD\kappa_A}{\varepsilon_0}. \tag{39}
$$

**Part 3: Bound $\mathcal{N}(\mathcal{V}, \rho_{\mathbf{Fin}}, \varepsilon_0)$.** It is straightforward that

$$
\mathcal{N}(\mathcal{V}, \rho_{\mathbf{Fin}}, \varepsilon_0) \le \left( \frac{3}{\varepsilon_0} \right)^{2R \sum_{d=2}^{D} p_d}.
$$

Following the same argument in Part 3 of the proof for Theorem 4, we have

$$\int_0^{\varepsilon_0} [\log \mathcal{N}(\mathcal{V}, \rho_{\text{Fin}}, t)]^{1/2} dt \lesssim \varepsilon_0 \sqrt{R \sum_{d=2}^D p_d \log \frac{1}{\varepsilon_0}}. \tag{40}$$

From Lemma 5, we have

$$\widetilde{\alpha}^2 = \max_{i \in [n]} \ell_i^2 \left( A^{\vartheta_{\backslash 1}, \varphi_{\backslash 1}} \right) \leq \max_{i \in [n]} \ell_i^2(A) \leq \frac{1}{\left( R \sum_{d=2}^D p_d \right)^2}. \tag{41}$$

Combining (38) – (41) and Theorem 3, we have that the claim holds if

$$m \gtrsim \varepsilon^{-2} R \left( p_1 + \sum_{d=2}^D p_d \cdot \log \frac{R D \kappa_A}{\varepsilon_0} \right) (\log^4 m)(\log^5 n),$$

$$s \gtrsim \varepsilon^{-2} \left( \log^2 \frac{1}{\varepsilon_0} + \varepsilon_0^2 R \sum_{d=1}^D p_d \log \frac{1}{\varepsilon_0} \right) (\log^6 m)(\log^5 n).$$

We finish the proof by taking $\varepsilon_0 = \frac{1}{R \sum_{d=1}^D p_d}$. Note that this choice of $\varepsilon$ satisfies the requirement in Part 2.

## G  Proof of Theorem 2

Denote $A^{\left\{ \vartheta_{\backslash 1}^{\{r_d\}}, \varphi_{\backslash 1}^{\{r_d\}} \right\}} = \left[ A^{\left\{ \theta_{\backslash 1}^{\{r_d\}} \right\}}, A^{\left\{ \phi_{\backslash 1}^{\{r_d\}} \right\}} \right]$. We illustrate that the set $\mathcal{T}$ can be reparameterized to the following set with respect to tensors with partial orthogonal factors:

$$\mathcal{T} = \bigcup_{E \in \mathcal{V}} \{ x \in E \mid \|x\|_2 = 1 \}, \quad \text{where } \mathcal{V} = \bigcup_{\widehat{\mathcal{W}}} \text{span} \left( A^{\left\{ \vartheta_{\backslash 1}^{\{r_d\}}, \varphi_{\backslash 1}^{\{r_d\}} \right\}} \right) \quad \text{and}$$

$$\widetilde{\mathcal{W}} = \Big\{ \forall r_d \in [R_d], d \in [D]\backslash\{1\}, \theta_d^{(r_d)}, \phi_d^{(r_d)} \in \mathcal{B}_{p_d}; \forall r_d, q_d \in [R_d], \exists d \in [D]\backslash\{1\} \text{ s.t. } \langle \theta_d^{(r_d)}, \phi_d^{(q_d)} \rangle = 0;$$

$$\forall r_d \in [R_d - 1], q_d \in [R_d]\backslash[r_d], \exists d, t \in [D]\backslash\{1\} \text{ s.t. } \langle \theta_d^{(r_d)}, \theta_d^{(q_d)} \rangle = \langle \phi_t^{(r_d)}, \phi_t^{(q_d)} \rangle = 0 \Big\}.$$

Repeating the argument in the proof of Theorem 1, we have the equivalence of $\mathcal{T}$ and the set above.

**Part 1: Bound $p_{\mathcal{V}}$.** It is straightforward that

$$p_{\mathcal{V}} = \sup_{\widetilde{\mathcal{W}}} \dim \left\{ \text{span} \left( A^{\left\{ \vartheta_{\backslash 1}^{\{r_d\}}, \varphi_{\backslash 1}^{\{r_d\}} \right\}} \right) \right\} \leq 2 R_1 p_1. \tag{42}$$

**Part 2: Bound $\gamma_2^2(\mathcal{V}, \rho_{\text{Fin}})$.** The $\gamma_2$-functional in this case is

$$\gamma_2^2(\mathcal{V}, \rho_{\text{Fin}}) = \inf_{\{\overline{\mathcal{V}}_k\}_{k=0}^\infty} \sup_{A^{\left\{ \vartheta_{\backslash 1}^{\{r_d\}}, \varphi_{\backslash 1}^{\{r_d\}} \right\}} \in \mathcal{V}} \sum_{k=0}^\infty 2^{r/2} \cdot \rho_{\text{Fin}} \left( A^{\left\{ \vartheta_{\backslash 1}^{\{r_d\}}, \varphi_{\backslash 1}^{\{r_d\}} \right\}}, \overline{\mathcal{V}}_k \right),$$

where $\overline{\mathcal{V}}_k$ is an $\varepsilon_k$-net of $\mathcal{V}$.

Following the same argument as in Part 2 of the proof for Theorem 4, we have from Lemma 10 that if $k'$ is the smallest integer such that $2\kappa_A \left( (1 + \eta_{k'})^D - 1 \right) \sqrt{\prod_{d=2}^D R_d} \leq 1$, then we choose $\varepsilon_0$ small enough such that

$$\varepsilon \leq 2 D \kappa_A \eta_{k'} \sqrt{\prod_{d=2}^D R_d} \leq 2 C \kappa_A \left( (1 + \eta_{k'})^D - 1 \right) R_1 \sqrt{\text{nnz}(G)},$$

where the second inequality follows from the binomial theorem. Then we have

$$\gamma_2^2(\mathcal{V}, \rho_{\text{Fin}}) \lesssim \left( \sum_{d=2}^{D} R_d p_d + \text{nnz}(G) \right) \cdot \log \frac{D \kappa_A \sqrt{\prod_{d=2}^{D} R_d}}{\varepsilon_0}. \tag{43}$$

**Part 3: Bound** $\mathcal{N}(\mathcal{V}, \rho_{\text{Fin}}, \varepsilon_0)$. It is straightforward that

$$\mathcal{N}(\mathcal{V}, \rho_{\text{Fin}}, \varepsilon_0) \le \left( \frac{3}{\varepsilon_0} \right)^{2 \left( \sum_{d=2}^{D} R_d p_d + \text{nnz}(G) \right)}.$$

Following the same argument in Part 3 of the proof for Theorem 4, we have

$$\int_0^{\varepsilon_0} [\log \mathcal{N}(\mathcal{V}, \rho_{\text{Fin}}, t)]^{1/2} dt \lesssim \varepsilon_0 \sqrt{\left( \sum_{d=2}^{D} R_d p_d + \text{nnz}(G) \right) \log \frac{1}{\varepsilon_0}}. \tag{44}$$

From Lemma 5, we have

$$\widetilde{\alpha}^2 = \max_{i \in [n]} \ell_i^2 \left( A^{\vartheta \setminus 1, \varphi \setminus 1} \right) \le \max_{i \in [n]} \ell_i^2(A) \le 1 / \left( \sum_{d=2}^{D} R_d p_d + \text{nnz}(G) \right)^2. \tag{45}$$

Combining (38) – (41) and Theorem 3, we have that the claim holds if

$$m \gtrsim \varepsilon^{-2} \left( R_1 p_1 + \left( \sum_{d=2}^{D} R_d p_d + \text{nnz}(G) \right) \cdot \log \frac{D \kappa_A R_1 \sqrt{\text{nnz}(G)}}{\varepsilon_0} \right) (\log^4 m)(\log^5 n),$$

$$s \gtrsim \varepsilon^{-2} \left( \log^2 \frac{1}{\varepsilon_0} + \varepsilon_0^2 \left( \sum_{d=1}^{D} R_d p_d + \text{nnz}(G) \right) \log \frac{1}{\varepsilon_0} \right) (\log^6 m)(\log^5 n).$$

We finish the proof by taking $\varepsilon_0 = \frac{1}{\sum_{d=1}^{D} R_d p_d + \text{nnz}(G)}$. Note that this choice of $\varepsilon$ satisfies the requirement in Part 2.

## H  Flattening Leverage Scores

Our analysis makes the weak assumption that the leverage scores of the design $A$ are slightly upper bounded. This might be restrictive if we have no control on the design $A$ at all. In the sequel, we apply a standard idea [9, 26] to flatten the leverage scores of a deterministic design $A$ based on the subsampled randomized hadamard transformation (SRHT) using the Walsh-Hadamard matrix. An SRHT matrix is defined as $\Psi = \sqrt{\frac{n}{m}} \Phi H \Sigma$, where the components $\Sigma$, $H$, and $\Phi$ are generated as:

(G1)  $\Sigma$ is an $n \times n$ diagonal matrix, where $\Sigma_{ii} = 1$ or -1 with equal probabilities 1/2.

(G2)  $H$ is an $n \times n$ orthogonal matrix generated from a Walsh-Hadamard matrix scaled by $n^{-1/2}$.

(G3)  $\Phi$ is an $m \times n$ SJLT matrix, with column sparsity bounded by $s$.

Note that computing a matrix-vector product with $H$ takes $\mathcal{O}(n \log n)$ instead of $n^2$ time. Thus, one can compute $H \Sigma A$ for an $n \times d$ matrix $A$ in $O(nd \log n)$ time, which is well-suited for the case in which $A$ is dense, e.g., $\text{nnz}(A) = \Theta(nd)$. The purpose of the matrix product $H \Sigma$ is to uniformize the leverage scores before applying our SJLT with $\Phi$.

We next give a standard lemma for flattening the leverage scores, included for completeness. Without loss of generality, we assume that $n = 2^q$ for a positive integer $q$, implying that a Walsh-Hadamard matrix exists.

**Lemma 1.** Suppose $H$ and $\Sigma$ are generated as in (G1) and (G2). Given any real value $\delta \in (0, 1)$ and an $n \times d$ matrix $A$ with $\text{rank}(A) = r$, with probability at least $1 - \delta$, we have

$$\max_{i \in [n]} \ell_i^2(H \Sigma A) \lesssim \frac{r \cdot \log \left( \frac{nr}{\delta} \right)}{n}.$$

*Proof.* Given a unit vector $y \in \mathbb{R}^n$, let $Z_{jk} = H_{jk}\Sigma_{kk}y_k$ for all $j \in [n]$. Then from the independence of $H_{jk}$ and $\Sigma_{kk}$, we have

$$\mathbb{E}(Z_{jk}) = \mathbb{E}(H_{jk}\Sigma_{kk}y_k) = \mathbb{E}(H_{jk}) \cdot \mathbb{E}(\Sigma_{kk}) \cdot y_k = 0,$$

$$\text{Var}(Z_{jk}) \leq \mathbb{E}(H_{jk}^2\Sigma_{kk}^2y_k^2) = \mathbb{E}(H_{jk}^2) \cdot \mathbb{E}(\Sigma_{kk}^2) \cdot y_k^2 = \frac{y_k^2}{n}.$$

From the Azuma-Hoeffding inequality, for any $t > 0$ we have

$$\mathbb{P}\left(\left|\sum_{k=1}^n Z_{jk}\right| > t\right) \leq 2\exp\left(-\frac{nt^2}{2\sum_{k=1}^n y_k^2}\right) = 2\exp\left(-\frac{nt^2}{2}\right).$$

By taking $t = \sqrt{\frac{2\log\left(\frac{2nr}{\delta}\right)}{n}}$, we have

$$\mathbb{P}\left(\left|\sum_{k=1}^n Z_{jk}\right| > \sqrt{\frac{2\log\left(\frac{2nr}{\delta}\right)}{n}}\right) \leq 2\exp\left(\log\left(\frac{\delta}{2nr}\right)\right) = \frac{\delta}{nr}.$$

By a union bound, we have

$$\mathbb{P}\left(\|H\Sigma y\|_\infty > \sqrt{\frac{2\log\left(\frac{2nr}{\delta}\right)}{n}}\right) = \mathbb{P}\left(\max_{j \in [n]}\left|\sum_{k=1}^n Z_{jk}\right| > \sqrt{\frac{2\log\left(\frac{2nr}{\delta}\right)}{n}}\right) \leq \frac{\delta}{r}.$$

Suppose $A = UQ$, where $U \in \mathbb{R}^{n \times r}$ has orthonormal columns. Then we have for all $i \in [n]$ and $k \in [r]$,

$$\ell_i^2(H\Sigma A) = \ell_i^2(H\Sigma U) \leq r \cdot \left(e_i^\top H\Sigma U e_k\right)^2.$$

Using a union bound again, we finish the proof by

$$\mathbb{P}\left(\max_{i \in [n]} \ell_i^2(H\Sigma A) > \frac{2r\log\left(\frac{2nr}{\delta}\right)}{n}\right) \leq \mathbb{P}\left(\max_{i \in [n]} r \cdot \left\|e_i^\top H\Sigma U e_k\right\|_\infty^2 > \frac{2r\log\left(\frac{2nr}{\delta}\right)}{n}\right) \leq \delta.$$

$\square$

Applying this with the bound $\max_{i \in [n]} \ell_i^2(H\Sigma A) \leq 1/(R \cdot \sum_{d=2}^D p_d)^2$ of Theorem 1 gives:

**Proposition 1.** Suppose $H$ and $\Sigma$ are generated as in (G1) and (G2). Denote $C_2 = R\sum_{d=2}^D p_d$. For low-rank tensor regression (4), where $A \in \mathbb{R}^{n \times \prod p_d}$ is the matricization of all tensor designs, if $n$ satisfies $n \gtrsim C_2^2 \cdot \text{rank}(A) \cdot \log\left(n \cdot \text{rank}(A)/\delta\right)$, then with probability at least $1 - \delta$, we have $\max_{i \in [n]} \ell_i^2(H\Sigma A) \leq 1/C_2^2$.

Combining Theorem 1 and Proposition 1, we achieve (8), provided $n$ is sufficiently large. Here we use that for all $x$, $\|H\Sigma Ax\|_2 = \|Ax\|_2$ since $H\Sigma$ is an isometry.

In the worst case, $\text{rank}(A) = \prod p_d$, which requires $n = \Omega\left(R^2\left(\sum_{d=2}^D p_d\right)^2 \cdot \prod p_d\right)$. In overconstrained regression, it is often assumed that the number $n$ of examples is at least a small polynomial in $\text{rank}(A)$ [30], which implies this bound on $n$. Also, if, for example, $A_i$ is sampled from a distribution with a rank deficient covariance, one may even have $\text{rank}(A) \ll \prod p_d$. A similar argument applies to the Tucker model as well in Theorem 2.

One should note that computing $\Phi H\Sigma A$ takes $(n\log n)\prod_{d=1}^D p_d$ time, provided the column sparsity $s$ of $\Phi$ is $O(1)$. This is $O(\text{nnz}(A)\log n)$ time for dense matrices $A$, i.e., those with $\text{nnz}(A) = \Omega(nd)$, but in general, unlike our earlier results, is not $O(\text{nnz}(A)\log n)$ time for sparse matrices. Analogous results can be obtained for the Tucker decomposition model, which we omit.

# I  Intermediate Results

Here we introduce all intermediate results applied in our main analysis.

**Lemma 2.** Suppose for $A = [A^{(1)}, A^{(2)}, \ldots, A^{(m)}] \in \mathbb{R}^{n \times mp}$, each $A^{(i)} \in \mathbb{R}^{n \times p}$ is a column-wise sub-matrix of $A$. Given a vector $v \in \mathbb{R}^m$, we have

$$\left\| \sum_{i=1}^{m} A^{(i)} v_i \right\|_2 \le \|A\|_2 \|v\|_2.$$

*Proof.* This is an extension of the Cauchy-Schwartz inequality. We have $\sum_{i=1}^{m} A^{(i)} v_i = A(v \otimes I_p)$, where $\otimes$ is the Kronecker product. This implies

$$\left\| \sum_{i=1}^{m} A^{(i)} v_i \right\|_2 = \|A(v \otimes I_p)\|_2 \le \|A\|_2 \|v \otimes I_p\|_2 = \|A\|_2 \|v\|_2.$$

$\square$

**Lemma 3.** Given two sequences of unit vectors $\{\phi_i\}_{i=1}^{n}$ and $\{\psi_i\}_{i=1}^{n}$, where $\phi_i, \psi_i \in \mathbb{R}^{p_i}$ with $\|\phi_i - \psi_i\|_2 \le \varepsilon$ for all $i \in [n]$, we have

$$\|\phi_1 \otimes \phi_2 \otimes \cdots \otimes \phi_n - \psi_1 \otimes \psi_2 \otimes \cdots \otimes \psi_n\|_2 \le (1 + \varepsilon)^n - 1.$$

*Proof.* Suppose for all $i \in [n]$, we have $\psi_i = \phi_1 + x_i$ for some vector $x_i \in \mathbb{R}^{p_i}$. Then we have

$$\|\phi_1 \otimes \cdots \otimes \phi_n - \psi_1 \otimes \cdots \otimes \psi_n\|_2 = \|\phi_1 \otimes \cdots \otimes \phi_n - (\phi_1 + x_i) \otimes \cdots \otimes (\psi_n + x_n)\|_2$$

$$\le \sum_{i=1}^{n} \|\phi_1 \otimes \cdots \otimes x_i \otimes \cdots \otimes \phi_n\|_2$$

$$+ \sum_{i=1}^{n} \sum_{j=1, j \ne i}^{n} \|\phi_1 \otimes \cdots \otimes x_i \otimes \cdots \otimes x_j \otimes \cdots \otimes \phi_n\|_2 + \cdots + \|x_1 \otimes \cdots \otimes x_n\|_2$$

$$\le \binom{n}{1} \varepsilon + \binom{n}{2} \varepsilon^2 + \cdots + \binom{n}{n} \varepsilon^n = (1 + \varepsilon)^n - 1,$$

where the last inequality is from the fact that $\|v \otimes u\|_2 = \|v\|_2 \|u\|_2$ for any vectors $v$ and $u$. $\square$

**Lemma 4.** Suppose that $A \in \mathbb{R}^{n \times \prod_{d=1}^{2} p_d}$ has leverage scores $\ell_i^2(A)$ for all $i \in [n]$. Then for any $v_1, v_2 \in \mathbb{R}^{p_2}$, the leverage scores of $A^{v_1, v_2} = [A^{v_1}, A^{v_2}] \in \mathbb{R}^{n \times 2p_1}$ are bounded by $\ell_i^2(A^{v_1, v_2}) \le \ell_i^2(A)$.

*Proof.* Let $Z$ have orthonormal columns and have the same span as the column space of $A$. Then we have $\ell_i^2(A) = \|e_i^\top Z\|_2^2$ for all $i \in [n]$. Since the column space of $A^{v_1, v_2}$ is a subspace of the column space of $A$, we can always find a column sub-matrix $Z_1 \in \mathbb{R}^{n \times 2p_1}$ of $Z$ such that $Z_1$ spans the column space of $A^{v_1, v_2}$. Therefore, for each $i \in [n]$, we have

$$\ell_i^2(A^{v_1, v_2}) = \|e_i^\top Z_1\|_2^2 \le \|e_i^\top Z\|_2^2 = \ell_i^2(A).$$

$\square$

**Lemma 5.** Suppose $A \in \mathbb{R}^{n \times \prod_{d=1}^{2} p_d}$ has leverage scores $\ell_i^2(A)$ for all $i \in [n]$. Then for any $v_i^{(r)} \in \mathbb{R}^{p_2}$, $i \in [2]$, $r \in [R]$ with $R \le p_2/2$, the leverage scores of $A^{\left\{ v_i^{(r)} \right\}} = \left[ A^{v_1^{(1)}}, \ldots, A^{v_1^{(R)}}, A^{v_2^{(1)}}, \ldots, A^{v_2^{(R)}} \right] \in \mathbb{R}^{n \times 2Rp_1}$ are bounded by $\ell_i^2 \left( A^{\left\{ v_i^{(r)} \right\}} \right) \le \ell_i^2(A)$.

*Proof.* Let $Z$ have orthonormal columns and have the same span as the column space of $A$. Then we have $\ell_i^2(A) = \|e_i^\top Z\|_2^2$ for all $i \in [n]$. Since the column space of $A^{\left\{ v_i^{(r)} \right\}}$ is a subspace of the column space of $A$, as the column space of each $A^{v_i^{(r)}}$ is a subspace of the column space of $A$, we can always find a column sub-matrix $Z_1 \in \mathbb{R}^{n \times 2Rp_1}$ of $Z$ such that $Z_1$ spans the column space of $A^{\left\{ v_i^{(r)} \right\}}$. Therefore, for each $i \in [n]$, we have

$$\ell_i^2 \left( A^{\left\{ v_i^{(r)} \right\}} \right) = \|e_i^\top Z_1\|_2^2 \le \|e_i^\top Z\|_2^2 = \ell_i^2(A).$$

$\square$

**Lemma 6.** For any $v_1, v_2 \in \mathcal{B}_{p_2}$, suppose $\langle v_1, v_2 \rangle = 0$, and $\overline{v}_1, \overline{v}_2 \in \mathcal{B}_{p_2}$ are vectors such that $\|v_1 - \overline{v}_1\|_2 \leq \eta_0$ and $\|v_2 - \overline{v}_2\|_2 \leq \eta_0$. Then we have

$$\rho_{\mathrm{Fin}}([A^{v_1}, A^{v_2}], [A^{\overline{v}_1}, A^{\overline{v}_2}]) \leq 2\kappa_A \eta_0.$$

*Proof.* Denote $A^{v_1, v_2} = [A^{v_1}, A^{v_2}]$. From a perturbation bound for orthogonal projections given in [14], we have

$$\rho_{\mathrm{Fin}}(A^{v_1, v_2}, A^{\overline{v}_1, \overline{v}_2}) \leq \frac{\|A^{v_1, v_2} - A^{\overline{v}_1, \overline{v}_2}\|_2}{\sigma_{\min}(A^{v_1, v_2})}. \tag{46}$$

We first provide an upper bound on the numerator as

$$\|A^{v_1, v_2} - A^{\overline{v}_1, \overline{v}_2}\|_2 = \left\| \left[ \sum_{i=1}^{p_2} A^{(i)}(v_{1,i} - \overline{v}_{1,i}), \sum_{i=1}^{p_2} A^{(i)}(v_{2,i} - \overline{v}_{2,i}) \right] \right\|_2$$

$$\leq \left\| \sum_{i=1}^{p_2} A^{(i)}(v_{1,i} - \overline{v}_{1,i}) \right\|_2 + \left\| \sum_{i=1}^{p_2} A^{(i)}(v_{2,i} - \overline{v}_{2,i}) \right\|_2$$

$$\leq 2\sigma_{\max}(A)\eta_0, \tag{47}$$

where the last inequality is from Lemma 2.

Next, we provide a lower bound on the denominator. Let $[u_1^\top, u_2^\top]^\top$ be a unit vector corresponding to the smallest singular value of $A^{v_1, v_2}$, where $u_1, u_2 \in \mathbb{R}^{p_1}$. Then we have

$$\sigma_{\min}(A^{v_1, v_2}) = \left\| A^{v_1, v_2} \begin{bmatrix} u_1 \\ u_2 \end{bmatrix} \right\|_2 = \|A(v_1 \otimes u_1 + v_2 \otimes u_2)\|_2 \geq \sigma_{\min}(A)\|v_1 \otimes u_1 + v_2 \otimes u_2\|_2$$

$$= \sigma_{\min}(A)\sqrt{\|v_1 \otimes u_1\|_2^2 + \|v_2 \otimes u_2\|_2^2 + 2\langle v_1 \otimes u_1, v_2 \otimes u_2 \rangle}$$

$$= \sigma_{\min}(A)\sqrt{\|u_1\|_2^2 + \|u_2\|_2^2 + 2\sum_{i=1}^{p_2}\sum_{j=1}^{p_1} v_{1,i}u_{1,j}v_{2,i}u_{2,j}}$$

$$= \sigma_{\min}(A)\sqrt{1 + 2\langle v_1, v_2 \rangle\langle u_1, u_2 \rangle} = \sigma_{\min}(A), \tag{48}$$

where the last equality is from the condition $\langle v_1, v_2 \rangle = 0$. We finish the proof by combining (46), (47), and (48). $\square$

**Lemma 7.** For all $i \in [2]$ and $r \in [R]$, $v_i^{(r)} \in \mathcal{B}_{p_2}$. Suppose for all $i \in [2]$, $r \in [R]$, $q \in [R]\backslash\{r\}$, we have $\langle v_i^{(r)}, v_i^{(q)} \rangle = \langle v_1^{(r)}, v_2^{(r)} \rangle = 0$. Further suppose for all $i \in [2]$ and $r \in [R]$, $\overline{v}_i^{(r)} \in \mathcal{B}_{p_2}$ is a vector such that $\|v_i^{(r)} - \overline{v}_i^{(r)}\|_2 \leq \eta_0$. Denote $A^{\{v_i^{(r)}\}} = \left[ A^{v_1^{(1)}}, \ldots, A^{v_1^{(R)}}, A^{v_2^{(1)}}, \ldots, A^{v_2^{(R)}} \right]$. Then we have

$$\rho_{\mathrm{Fin}}\left( A^{\{v_i^{(r)}\}}, A^{\{\overline{v}_i^{(r)}\}} \right) \leq 2R\kappa_A \eta_0.$$

*Proof.* From the perturbation bound for orthogonal projection given in [14], we have

$$\rho_{\mathrm{Fin}}\left( A^{\{v_i^{(r)}\}}, A^{\{\overline{v}_i^{(r)}\}} \right) \leq \frac{\left\| A^{\{v_i^{(r)}\}} - A^{\{\overline{v}_i^{(r)}\}} \right\|_2}{\sigma_{\min}\left( A^{\{v_i^{(r)}\}} \right)}. \tag{49}$$

We first upper bound the numerator as

$$
\left\| A^{\{v_i^{(r)}\}} - A^{\{\overline{v}_i^{(r)}\}} \right\|_2 = \left\| \left[ \sum_{j=1}^{p_2} A_j \left( v_{1,j}^{(1)} - \overline{v}_{1,j}^{(1)} \right), \ldots, \sum_{j=1}^{p_2} A_j \left( v_{1,j}^{(R)} - \overline{v}_{1,j}^{(R)} \right), \right. \right.
$$

$$
\left. \left. \sum_{j=1}^{p_2} A_j \left( v_{2,j}^{(1)} - \overline{v}_{2,j}^{(1)} \right), \ldots, \sum_{j=1}^{p_2} A_j \left( v_{2,j}^{(R} - \overline{v}_{2,j}^{(R)} \right) \right] \right\|_2
$$

$$
\leq \sum_{r=1}^{R} \left\| \sum_{j=1}^{p_2} A_j \left( v_{1,j}^{(r)} - \overline{v}_{1,j}^{(r)} \right) \right\| + \sum_{r=1}^{R} \left\| \sum_{j=1}^{p_2} A_j \left( v_{2,j}^{(r)} - \overline{v}_{2,j}^{(r)} \right) \right\|_2 \leq 2R\sigma_{\max}(A)\eta_0, \tag{50}
$$

where the last inequality is from Lemma 2.

Next, we provide a lower bound on the denominator. Let $\left[ u_1^{(1)\top}, \ldots, u_1^{(R)\top}, u_2^{(1)\top}, \ldots, u_2^{(R)\top} \right]^{\top} \in$ $\mathbb{R}^{2Rp_1}$ be a unit vector corresponding to the smallest singular value of $A^{\{v_i^{(r)}\}}$, where $u_i^{(r)} \in \mathbb{R}^{p_1}$ for all $i \in [2]$ and $r \in [R]$. Then we have

$$
\sigma_{\min}\left( A^{\{v_i^{(r)}\}} \right) = \left\| A^{\{v_i^{(r)}\}} \left[ u_1^{(1)\top}, \ldots, u_1^{(R)\top}, u_2^{(1)\top}, \ldots, u_2^{(R)\top} \right]^{\top} \right\|_2
$$

$$
= \left\| A \cdot \left( \sum_{r=1}^{R} v_1^{(r)} \otimes u_1^{(r)} + v_2^{(r)} \otimes u_2^{(r)} \right) \right\|_2 \geq \sigma_{\min}(A) \left\| \sum_{r=1}^{R} \left( v_1^{(r)} \otimes u_1^{(r)} + v_2^{(r)} \otimes u_2^{(r)} \right) \right\|_2
$$

$$
= \sigma_{\min}(A) \sqrt{ \sum_{r=1}^{R} \left( \left\| u_1^{(r)} \right\|_2^2 + \left\| u_2^{(r)} \right\|_2^2 \right) + 2 \sum_{r=1}^{R} \sum_{j=1}^{p_2} \sum_{k=1}^{p_1} v_{1,j}^{(r)} u_{1,k}^{(r)} v_{2,j}^{(r)} u_{2,k}^{(r)} }
$$

$$
\overline{ +2 \sum_{i=1}^{2} \sum_{r=1}^{R-1} \sum_{q=r+1}^{R} \sum_{j=1}^{p_2} \sum_{k=1}^{p_1} v_{i,j}^{(r)} u_{i,k}^{(r)} v_{i,j}^{(q)} u_{i,k}^{(q)} }
$$

$$
= \sigma_{\min}(A) \sqrt{ 1 + 2 \sum_{r=1}^{R} \langle v_1^{(r)}, v_2^{(r)} \rangle \langle u_1^{(r)}, u_2^{(r)} \rangle + 2 \sum_{i=1}^{2} \sum_{r=1}^{R-1} \sum_{q=r+1}^{R} \langle v_i^{(r)}, v_i^{(q)} \rangle \langle u_i^{(r)}, u_i^{(q)} \rangle }
$$

$$
= \sigma_{\min}(A), \tag{51}
$$

where the last equality uses the conditions that for all $i \in [2]$ and $r \in [R]$, $\langle v_i^{(r)}, v_i^{(q)} \rangle = \langle v_1^{(r)}, v_2^{(r)} \rangle = 0$ for $q \in [R] \backslash \{r\}$. We finish the proof by combining (49), (50), and (51). $\square$

**Lemma 8.** For all $d \in [D] \backslash \{1\}$, $\theta_d, \phi_d \in \mathcal{B}_{p_d}$. Suppose there exists an $i \in [D] \backslash \{1\}$ such that $\langle \theta_i, \phi_i \rangle = 0$. Further suppose for all $d \in [D] \backslash \{1\}$, $\overline{\theta}_d, \overline{\phi}_d \in \mathcal{B}_{p_d}$ are vectors such that $\|\theta_d - \overline{\theta}_d\|_2 \leq \eta_0$ and $\|\phi_d - \overline{\phi}_d\|_2 \leq \eta_0$. Then we have

$$
\rho_{\mathsf{Fin}}\left( \left[ A^{\{\theta_{\backslash 1}\}}, A^{\{\phi_{\backslash 1}\}} \right], \left[ A^{\{\overline{\theta}_{\backslash 1}\}}, A^{\{\overline{\phi}_{\backslash 1}\}} \right] \right) \leq 2\kappa_A \left( (1 + \eta_0)^{D-1} - 1 \right).
$$

*Proof.* Let $A^{\vartheta_{\backslash 1}, \varphi_{\backslash 1}} = \left[ A^{\{\theta_{\backslash 1}\}}, A^{\{\phi_{\backslash 1}\}} \right] \in \mathbb{R}^{n \times 2p_1}$. From the perturbation bound for orthogonal projection given in [14], we have

$$
\rho_{\mathsf{Fin}}\left( A^{\vartheta_{\backslash 1}, \varphi_{\backslash 1}}, A^{\overline{\vartheta}, \overline{\varphi}} \right) \leq \frac{ \left\| A^{\vartheta_{\backslash 1}, \varphi_{\backslash 1}} - A^{\overline{\vartheta}, \overline{\varphi}} \right\|_2 }{ \sigma_{\min}(A^{\vartheta_{\backslash 1}, \varphi_{\backslash 1}}) }. \tag{52}
$$

We denote $\sum_{j_2\cdots j_D} = \sum_{j_D=1}^{p_D}\cdots\sum_{j_2=1}^{p_2}$. We first provide an upper bound on the numerator:

$$\left\|A^{\vartheta\backslash 1,\varphi\backslash 1} - A^{\overline{\vartheta},\overline{\varphi}}\right\|_2$$

$$= \left\|\left[\sum_{j_2\cdots j_D} A^{(j_D,\ldots,j_2)}\left(\theta_{D,j_D}\cdots\theta_{2,j_2} - \overline{\theta}_{D,j_D}\cdots\overline{\theta}_{2,j_2}\right), \sum_{j_2\cdots j_D} A^{(j_D,\ldots,j_2)}\left(\phi_{D,j_D}\cdots\phi_{2,j_2} - \overline{\phi}_{D,j_D}\cdots\overline{\phi}_{2,j_2}\right)\right]\right\|_2$$

$$\le \left\|\sum_{j_2\cdots j_D} A^{(j_D,\ldots,j_2)}\cdot\left(\theta_{D,j_D}\cdots\theta_{2,j_2} - \overline{\theta}_{D,j_D}\cdots\overline{\theta}_{2,j_2}\right)\right\|_2 + \left\|\sum_{j_2\cdots j_D} A^{(j_D,\ldots,j_2)}\cdot\left(\phi_{D,j_D}\cdots\phi_{2,j_2} - \overline{\phi}_{D,j_D}\cdots\overline{\phi}_{2,j_2}\right)\right\|_2$$

$$\le \sigma_{\max}(A)\cdot\left(\left\|\theta_D\otimes\cdots\otimes\theta_2 - \overline{\theta}_D\otimes\cdots\otimes\overline{\theta}_2\right\|_2 + \left\|\phi_D\otimes\cdots\otimes\phi_2 - \overline{\phi}_D\otimes\cdots\otimes\overline{\phi}_2\right\|_2\right)$$

$$\le 2\sigma_{\max}(A)\left((1+\eta_0)^{D-1} - 1\right), \tag{53}$$

where the second inequality is from Lemma 2 and the last inequality is from Lemma 3.

Next, we provide a lower bound on the denominator. Let $[u_1^\top, u_2^\top]^\top$ be a unit vector corresponding to the smallest singular value of $A^{\vartheta\backslash 1,\varphi\backslash 1}$, where $u_1, u_2 \in \mathbb{R}^{p_1}$. Then we have

$$\sigma_{\min}\left(A^{\vartheta\backslash 1,\varphi\backslash 1}\right) = \left\|A^{\vartheta\backslash 1,\varphi\backslash 1}\begin{bmatrix}u_1\\u_2\end{bmatrix}\right\|_2 = \left\|A\left(\theta_D\otimes\cdots\otimes\theta_2\otimes u_1 + \phi_D\otimes\cdots\otimes\phi_2\otimes u_2\right)\right\|_2$$

$$\ge \sigma_{\min}(A)\|\theta_D\otimes\cdots\otimes\theta_2\otimes u_1 + \phi_D\otimes\cdots\otimes\phi_2\otimes u_2\|_2$$

$$= \sigma_{\min}(A)\sqrt{\|\theta_D\otimes\cdots\otimes\theta_2\otimes u_1\|_2^2 + \|\phi_D\otimes\cdots\otimes\phi_2\otimes u_2\|_2^2}$$

$$\overline{+2\langle\theta_D\otimes\cdots\otimes\theta_2\otimes u_1, \phi_D\otimes\cdots\otimes\phi_2\otimes u_2\rangle}$$

$$= \sigma_{\min}(A)\sqrt{\|u_1\|_2^2 + \|u_2\|_2^2 + 2\sum_{j_2\cdots j_D}\sum_{j_1=1}^{p_1}\theta_{D,j_D}\cdots\theta_{2,j_2}u_{1,j_1}\cdot\phi_{D,j_D}\cdots\phi_{2,j_2}u_{2,j_1}}$$

$$= \sigma_{\min}(A)\sqrt{1 + 2\langle\theta_D,\phi_D\rangle\cdots\langle\theta_2,\phi_2\rangle\langle u_1,u_2\rangle} = \sigma_{\min}(A), \tag{54}$$

where the last inequality is from $\langle\theta_i,\phi_i\rangle = 0$ for some $i \in \{2,\ldots,D\}$. We finish the proof by combining (52), (53) and (54). $\qquad\square$

**Lemma 9.** For all $d \in [D]\backslash\{1\}$ and $r \in [R]$, $\theta_d^{(r)}, \phi_d^{(r)} \in \mathcal{B}_{p_d}$. Suppose that for any $r, q \in [R]$, there exists an $i \in [D]\backslash\{1\}$ such that $\langle\theta_i^{(r)}, \phi_i^{(q)}\rangle = 0$, and further, for all $r \in [R-1]$, $q \in [R]\backslash[r]$, there exist $j, k \in [D]\backslash\{1\}$ such that $\langle\theta_j^{(r)}, \theta_j^{(q)}\rangle = 0$ and $\langle\phi_k^{(r)}, \phi_k^{(q)}\rangle = 0$. Further suppose for all $d \in [D]\backslash\{1\}$ and $r \in [R]$, $\overline{\theta}_d^{(r)}, \overline{\phi}_d^{(r)} \in \mathcal{B}_{p_d}$ are vectors such that $\|\theta_d^{(r)} - \overline{\theta}_d^{(r)}\|_2 \le \eta_0$ and $\|\phi_d^{(r)} - \overline{\phi}_d^{(r)}\|_2 \le \eta_0$. Then we have

$$\rho_{\text{Fin}}\left(\left[A^{\{\theta_{\backslash 1}^{(r)}\}}, A^{\{\phi_{\backslash 1}^{(r)}\}}\right], \left[A^{\{\overline{\theta}_{\backslash 1}^{(r)}\}}, A^{\{\overline{\phi}_{\backslash 1}^{(r)}\}}\right]\right) \le 2R\kappa_A\left((1+\eta_0)^{D-1} - 1\right).$$

*Proof.* Denote $A^{\{\vartheta_{\backslash 1}^{(r)},\varphi_{\backslash 1}^{(r)}\}} = \left[A^{\{\theta_{\backslash 1}^{(r)}\}}, A^{\{\phi_{\backslash 1}^{(r)}\}}\right] \in \mathbb{R}^{n\times 2Rp_1}$. From the perturbation bound on orthogonal projection given in [14], we have

$$\rho_{\text{Fin}}\left(A^{\{\vartheta_{\backslash 1}^{(r)},\varphi_{\backslash 1}^{(r)}\}}, A^{\{\overline{\vartheta}_{\backslash 1}^{(r)},\overline{\varphi}_{\backslash 1}^{(r)}\}}\right) \le \frac{\left\|A^{\{\vartheta_{\backslash 1}^{(r)},\varphi_{\backslash 1}^{(r)}\}} - A^{\{\overline{\vartheta}_{\backslash 1}^{(r)},\overline{\varphi}_{\backslash 1}^{(r)}\}}\right\|_2}{\sigma_{\min}\left(A^{\{\vartheta_{\backslash 1}^{(r)},\varphi_{\backslash 1}^{(r)}\}}\right)}. \tag{55}$$

We denote $\sum_{j_2\cdots j_D} = \sum_{j_D=1}^{p_D}\cdots\sum_{j_2=1}^{p_2}$. We first upper bound the numerator as

$$\left\|A^{\left\{\vartheta_{\backslash 1}^{(r)},\varphi_{\backslash 1}^{(r)}\right\}} - A^{\left\{\overline{\vartheta}_{\backslash 1}^{(r)},\overline{\varphi}_{\backslash 1}^{(r)}\right\}}\right\|_2$$

$$= \left\|\left[\sum_{j_2\cdots j_D} A^{(j_D,\ldots,j_2)}\left(\theta_{D,j_D}^{(1)}\cdots\theta_{2,j_2}^{(1)} - \overline{\theta}_{D,j_D}^{(1)}\cdots\overline{\theta}_{2,j_2}^{(1)}\right),\ldots,\sum_{j_2\cdots j_D} A^{(j_D,\ldots,j_2)}\left(\theta_{D,j_D}^{(R)}\cdots\theta_{2,j_2}^{(R)} - \overline{\theta}_{D,j_D}^{(R)}\cdots\overline{\theta}_{2,j_2}^{(R)}\right),\right.\right.$$

$$\left.\left.\sum_{j_2\cdots j_D} A^{(j_D,\ldots,j_2)}\left(\phi_{D,j_D}^{(1)}\cdots\phi_{2,j_2}^{(1)} - \overline{\phi}_{D,j_D}^{(1)}\cdots\overline{\phi}_{2,j_2}^{(1)}\right),\ldots,\sum_{j_2\cdots j_D} A^{(j_D,\ldots,j_2)}\left(\phi_{D,j_D}^{(R)}\cdots\phi_{2,j_2}^{(R)} - \overline{\phi}_{D,j_D}^{(R)}\cdots\overline{\phi}_{2,j_2}^{(R)}\right)\right]\right\|_2$$

$$\le \sum_{r=1}^{R}\left\|\sum_{j_2\cdots j_D} A^{(j_D,\ldots,j_2)}\cdot\left(\theta_{D,j_D}^{(r)}\cdots\theta_{2,j_2}^{(r)} - \overline{\theta}_{D,j_D}^{(r)}\cdots\overline{\theta}_{2,j_2}^{(r)}\right)\right\|_2 + \left\|\sum_{j_2\cdots j_D} A^{(j_D,\ldots,j_2)}\cdot\left(\phi_{D,j_D}^{(r)}\cdots\phi_{2,j_2}^{(r)} - \overline{\phi}_{D,j_D}^{(r)}\cdots\overline{\phi}_{2,j_2}^{(r)}\right)\right\|_2$$

$$\le \sigma_{\max}(A)\cdot\left(\sum_{r=1}^{R}\left\|\theta_D^{(r)}\otimes\cdots\otimes\theta_2^{(r)} - \overline{\theta}_D^{(r)}\otimes\cdots\otimes\overline{\theta}_2^{(r)}\right\|_2 + \left\|\phi_D^{(r)}\otimes\cdots\otimes\phi_2^{(r)} - \overline{\phi}_D^{(r)}\otimes\cdots\otimes\overline{\phi}_2^{(r)}\right\|_2\right)$$

$$\le 2R\sigma_{\max}(A)\left((1+\eta_0)^{D-1}-1\right),\tag{56}$$

where the second inequality is from Lemma 2 and the last inequality is from Lemma 3.

Next, we lower bound the denominator. Let $\left[u_1^{(1)\top},\ldots,u_1^{(R)\top},u_2^{(1)\top},\ldots,u_2^{(R)\top}\right]^\top \in \mathbb{R}^{2Rp_1}$ be a unit vector corresponding to the smallest singular value of $A^{\left\{\vartheta_{\backslash 1}^{(r)},\varphi_{\backslash 1}^{(r)}\right\}}$, where $u_i^{(r)} \in \mathbb{R}^{p_1}$ for all $i \in [2]$ and $r \in [R]$. Then we have

$$\sigma_{\min}\left(A^{\left\{\vartheta_{\backslash 1}^{(r)},\varphi_{\backslash 1}^{(r)}\right\}}\right) = \left\|A^{\left\{\vartheta_{\backslash 1}^{(r)},\varphi_{\backslash 1}^{(r)}\right\}}\left[u_1^{(1)\top},\ldots,u_1^{(R)\top},u_2^{(1)\top},\ldots,u_2^{(R)\top}\right]^\top\right\|_2$$

$$= \left\|A\cdot\left(\sum_{r=1}^{R}\theta_D^{(r)}\otimes\cdots\otimes\theta_2^{(r)}\otimes u_1^{(r)} + \phi_D^{(r)}\otimes\cdots\otimes\phi_2^{(r)}\otimes u_2^{(r)}\right)\right\|_2$$

$$\ge \sigma_{\min}(A)\left\|\sum_{r=1}^{R}\theta_D^{(r)}\otimes\cdots\otimes\theta_2^{(r)}\otimes u_1^{(r)} + \phi_D^{(r)}\otimes\cdots\otimes\phi_2^{(r)}\otimes u_2^{(r)}\right\|_2$$

$$= \sigma_{\min}(A)\sqrt{\begin{aligned}&\sum_{r=1}^{R}\left(\left\|u_1^{(r)}\right\|_2^2 + \left\|u_2^{(r)}\right\|_2^2\right) + 2\sum_{r=1}^{R}\sum_{q=1}^{R}\sum_{j_1\cdots j_D}\theta_{D,j_D}^{(r)}\cdots\theta_{2,j_2}^{(r)}u_{1,j_1}^{(r)}\cdot\phi_{D,j_D}^{(q)}\cdots\phi_{2,j_2}^{(q)}u_{2,j_1}^{(q)}\\ &+2\sum_{r=1}^{R-1}\sum_{q=r+1}^{R}\sum_{j_1\cdots j_D}\left(\theta_{D,j_D}^{(r)}\cdots\theta_{2,j_2}^{(r)}u_{1,j_1}^{(r)}\cdot\theta_{D,j_D}^{(q)}\cdots\theta_{2,j_2}^{(q)}u_{1,j_1}^{(q)} + \phi_{D,j_D}^{(r)}\cdots\phi_{2,j_2}^{(r)}u_{2,j_1}^{(r)}\cdot\phi_{D,j_D}^{(q)}\cdots\phi_{2,j_2}^{(q)}u_{2,j_1}^{(q)}\right)\end{aligned}}$$

$$= \sigma_{\min}(A)\sqrt{\begin{aligned}&1 + 2\sum_{r=1}^{R}\sum_{q=1}^{R}\langle\theta_D^{(r)},\phi_D^{(q)}\rangle\cdots\langle\theta_2^{(r)},\phi_2^{(q)}\rangle\langle u_1^{(r)},u_2^{(q)}\rangle\\ &+2\sum_{r=1}^{R-1}\sum_{q=r+1}^{R}\left(\langle\theta_D^{(r)},\theta_D^{(q)}\rangle\cdots\langle\theta_2^{(r)},\theta_2^{(q)}\rangle\langle u_1^{(r)},u_1^{(q)}\rangle + \langle\phi_D^{(r)},\phi_D^{(q)}\rangle\cdots\langle\phi_2^{(r)},\phi_2^{(q)}\rangle\langle u_2^{(r)},u_2^{(q)}\rangle\right)\end{aligned}}$$

$$= \sigma_{\min}(A),\tag{57}$$

where the last inequality is from the conditions on $\theta_d^{(r)}$ and $\phi_d^{(r)}$. We finish the proof by combining (55), (56), and (57). $\qquad\square$

**Lemma 10.** For all $d \in [D]\backslash\{1\}$ and $r \in [R]$, $\theta_d^{(r_d)},\phi_d^{(r_d)} \in \mathcal{B}_{p_d}$. Suppose that for any $r_d,q_d \in [R_d]$, $d \in [R]\backslash\{1\}$, there exists an $i \in [D]\backslash\{1\}$ such that $\langle\theta_i^{(r)},\phi_i^{(q)}\rangle = 0$, and for all $r \in [R-1]$, $q \in [R]\backslash[r]$, there exist $j,k \in [D]\backslash\{1\}$ such that $\langle\theta_j^{(r)},\theta_j^{(q)}\rangle = 0$ and $\langle\phi_k^{(r)},\phi_k^{(q)}\rangle = 0$. Further suppose for all $d \in [D]\backslash\{1\}$ and $r \in [R]$, $\overline{\theta}_d^{(r)},\overline{\phi}_d^{(r)} \in \mathcal{B}_{p_d}$ are vectors such that $\|\theta_d^{(r)} - \overline{\theta}_d^{(r)}\|_2 \le \eta_0$ and $\|\phi_d^{(r)} - \overline{\phi}_d^{(r)}\|_2 \le \eta_0$. Then for some constant $C$, we have

$$\rho_{\text{Fin}}\left(\left[A^{\left\{\theta_{\backslash 1}^{(r)}\right\}},A^{\left\{\phi_{\backslash 1}^{(r)}\right\}}\right],\left[A^{\left\{\overline{\theta}_{\backslash 1}^{(r)}\right\}},A^{\left\{\overline{\phi}_{\backslash 1}^{(r)}\right\}}\right]\right) \le 2C\kappa_A\left((1+\eta_0)^{D-1}-1\right)R_1\sqrt{nnz(G)}.$$

*Proof.* Denote $A^{\left\{\vartheta_{\backslash 1}^{\{r_d\}},\varphi_{\backslash 1}^{\{r_d\}}\right\}} = \left[A^{\left\{\theta_{\backslash 1}^{\{r_d\}}\right\}}, A^{\left\{\phi_{\backslash 1}^{\{r_d\}}\right\}}\right] \in \mathbb{R}^{n \times 2R_1 p_1}$. From the perturbation bound for orthogonal projection given in [14], we have

$$\rho_{\mathrm{Fin}}\left(A^{\left\{\vartheta_{\backslash 1}^{\{r_d\}},\varphi_{\backslash 1}^{\{r_d\}}\right\}}, A^{\left\{\overline{\vartheta}_{\backslash 1}^{\{r_d\}},\overline{\varphi}_{\backslash 1}^{\{r_d\}}\right\}}\right) \leq \frac{\left\|A^{\left\{\vartheta_{\backslash 1}^{\{r_d\}},\varphi_{\backslash 1}^{\{r_d\}}\right\}} - A^{\left\{\overline{\vartheta}_{\backslash 1}^{\{r_d\}},\overline{\varphi}_{\backslash 1}^{\{r_d\}}\right\}}\right\|_2}{\sigma_{\min}\left(A^{\left\{\vartheta_{\backslash 1}^{\{r_d\}},\varphi_{\backslash 1}^{\{r_d\}}\right\}}\right)}. \qquad (58)$$

We denote $\sum_{j_2 \cdots j_D} = \sum_{j_2=1}^{p_2} \cdots \sum_{j_D=1}^{p_D}$, $\sum_{r_2 \cdots r_D} = \sum_{r_2=1}^{R_2} \cdots \sum_{r_D=1}^{R_D}$, and $\sum_{r_1 \cdots r_D} = \sum_{r_1=1}^{R_1} \cdots \sum_{r_D=1}^{R_D}$. We first upper bound the numerator as

$$\left\|A^{\left\{\vartheta_{\backslash 1}^{\{r_d\}},\varphi_{\backslash 1}^{\{r_d\}}\right\}} - A^{\left\{\overline{\vartheta}_{\backslash 1}^{\{r_d\}},\overline{\varphi}_{\backslash 1}^{\{r_d\}}\right\}}\right\|_2$$

$$= \left\|\left[\sum_{r_2 \cdots r_D}\sum_{j_2 \cdots j_D} G(1, r_2, \ldots, r_D)A^{(j_D,\ldots,j_2)}\left(\theta_{D,j_D}^{(r_D)}\cdots\theta_{2,j_2}^{(r_2)} - \overline{\theta}_{D,j_D}^{(r_D)}\cdots\overline{\theta}_{2,j_2}^{(r_2)}\right), \ldots,\right.\right.$$

$$\sum_{r_2 \cdots r_D}\sum_{j_2 \cdots j_D} G(R_1, r_2, \ldots, r_D)A^{(j_D,\ldots,j_2)}\left(\theta_{D,j_D}^{(r_D)}\cdots\theta_{2,j_2}^{(r_2)} - \overline{\theta}_{D,j_D}^{(r_D)}\cdots\overline{\theta}_{2,j_2}^{(r_2)}\right),$$

$$\sum_{r_2 \cdots r_D}\sum_{j_2 \cdots j_D} G(1, r_2, \ldots, r_D)A^{(j_D,\ldots,j_2)}\left(\phi_{D,j_D}^{(r_D)}\cdots\phi_{2,j_2}^{(r_2)} - \overline{\phi}_{D,j_D}^{(r_D)}\cdots\overline{\phi}_{2,j_2}^{(r_2)}\right), \ldots,$$

$$\left.\left.\sum_{r_2 \cdots r_D}\sum_{j_2 \cdots j_D} G(R_1, r_2, \ldots, r_D)A^{(j_D,\ldots,j_2)}\left(\phi_{D,j_D}^{(r_D)}\cdots\phi_{2,j_2}^{(r_2)} - \overline{\phi}_{D,j_D}^{(r_D)}\cdots\overline{\phi}_{2,j_2}^{(r_2)}\right)\right]\right\|_2$$

$$\leq \sum_{r_1 \cdots r_D}\left\|\sum_{j_2 \cdots j_D} G(r_1, \ldots, r_D)A^{(j_D,\ldots,j_2)}\left(\theta_{D,j_D}^{(r_D)}\cdots\theta_{2,j_2}^{(r_2)} - \overline{\theta}_{D,j_D}^{(r_D)}\cdots\overline{\theta}_{2,j_2}^{(r_2)}\right)\right\|_2$$

$$+ \left\|\sum_{j_2 \cdots j_D} G(r_1, \ldots, r_D)A^{(j_D,\ldots,j_2)} \cdot \left(\phi_{D,j_D}^{(r_D)}\cdots\phi_{2,j_2}^{(r_2)} - \overline{\phi}_{D,j_D}^{(r_D)}\cdots\overline{\phi}_{2,j_2}^{(r_2)}\right)\right\|_2$$

$$\leq \sigma_{\max}(A)\left(\sum_{r_1 \cdots r_D}|G(r_1, \ldots, r_D)| \cdot \left\|\theta_D^{(r_D)}\otimes\cdots\otimes\theta_2^{(r_2)} - \overline{\theta}_D^{(r_D)}\otimes\cdots\otimes\overline{\theta}_2^{(r_2)}\right\|_2\right.$$

$$\left.+ |G(r_1, \ldots, r_D)| \cdot \left\|\phi_D^{(r_D)}\otimes\cdots\otimes\phi_2^{(r_2)} - \overline{\phi}_D^{(r_D)}\otimes\cdots\otimes\overline{\phi}_2^{(r_2)}\right\|_2\right)$$

$$\leq 2\|\mathrm{vec}(G)\|_1 \cdot \sigma_{\max}(A)\left((1+\eta_0)^{D-1} - 1\right), \qquad (59)$$

where the second inequality is from Lemma 2 and the last inequality is from Lemma 3.

Next, we provide a lower bound on the denominator. Let $\left[u_1^{(1)\top}, \ldots, u_1^{(R_1)\top}, u_2^{(1)\top}, \ldots, u_2^{(R_1)\top}\right]^\top \in \mathbb{R}^{2R_1 p_1}$ be a unit vector corresponding to the smallest singular value of $A^{\left\{\vartheta_{\backslash 1}^{\{r_d\}},\varphi_{\backslash 1}^{\{r_d\}}\right\}}$, where $u_i^{(r_1)} \in \mathbb{R}^{p_1}$ for all $i \in [2]$ and $r_1 \in [R_1]$. Denote $\sum_{j_1 \cdots j_D} = \sum_{j_1=1}^{p_1} \cdots \sum_{j_D=1}^{p_D}$, $\sum_{q_1 \cdots q_D} = \sum_{q_1=1}^{p_1} \cdots \sum_{q_D=1}^{p_D}$, $\sum_{r_d, j_d, q_d} = \sum_{r_1 \cdots r_D}\sum_{j_1 \cdots j_D}\sum_{q_1 \cdots q_D}$, $\mathbb{1}_{G(r_1,\ldots,r_D)\neq 0}$ as the indicator function that is 1 if $G(r_1, \ldots, r_D) \neq 0$ and is 0 otherwise, and $u_{\min}^{(r_1)} =$

$\min_{r_1}\left\{\left\|u_1^{(r_1)}\right\|_2^2+\left\|u_2^{(r_1)}\right\|_2^2\neq 0\right\}$. Then we have

$$\sigma_{\min}\left(A^{\left\{\vartheta_{\backslash 1}^{\{r_d\}},\varphi_{\backslash 1}^{\{r_d\}}\right\}}\right)=\left\|A^{\left\{\vartheta_{\backslash 1}^{\{r_d\}},\varphi_{\backslash 1}^{\{r_d\}}\right\}}\left[u_1^{(1)\top},\ldots,u_1^{(R_1)\top},u_2^{(1)\top},\ldots,u_2^{(R_1)\top}\right]^\top\right\|_2$$

$$=\left\|A\cdot\left(\sum_{r_1,\ldots,r_D}G(r_1,\ldots,r_D)\left(\theta_D^{(r_D)}\otimes\cdots\otimes\theta_2^{(r_2)}\otimes u_1^{(r_1)}+\phi_D^{(r_D)}\otimes\cdots\otimes\phi_2^{(r_2)}\otimes u_2^{(r_1)}\right)\right)\right\|_2$$

$$\geq\sigma_{\min}(A)\left\|\sum_{r_1,\ldots,r_D}G(r_1,\ldots,r_D)\left(\theta_D^{(r_D)}\otimes\cdots\otimes\theta_2^{(r_2)}\otimes u_1^{(r_1)}+\phi_D^{(r_D)}\otimes\cdots\otimes\phi_2^{(r_2)}\otimes u_2^{(r_1)}\right)\right\|_2$$

$$=\sigma_{\min}(A)\left(\sum_{r_1,\ldots,r_D}G^2(r_1,\ldots,r_D)\left(\left\|u_1^{(r_1)}\right\|_2^2+\left\|u_2^{(r_1)}\right\|_2^2\right)\right.$$

$$+2\sum_{r_d,j_d,q_d}G(r_1,\ldots,r_D)\left(\theta_{D,j_D}^{(r_D)}\cdots\theta_{2,j_2}^{(r_2)}u_{1,j_1}^{(r_1)}\phi_{D,j_D}^{(q_D)}\cdots\phi_{2,j_2}^{(q_2)}u_{2,j_1}^{(q_1)}\right.$$

$$\left.\left.+\theta_{D,j_D}^{(r_D)}\cdots\theta_{2,j_2}^{(r_2)}u_{1,j_1}^{(r_1)}\theta_{D,j_D}^{(q_D)}\cdots\theta_{2,j_2}^{(q_2)}u_{1,j_1}^{(q_1)}+\phi_{D,j_D}^{(r_D)}\cdots\phi_{2,j_2}^{(r_2)}u_{2,j_1}^{(r_1)}\phi_{D,j_D}^{(q_D)}\cdots\phi_{2,j_2}^{(q_2)}u_{2,j_1}^{(q_1)}\right)\right)^{1/2}$$

$$\geq\sigma_{\min}(A)\left(\min_{r_1}\sum_{r_2,\ldots,r_D}G^2(r_1,...,r_D)+2\sum_{r_d,j_d,q_d}G(r_1,...,r_D)\left(\langle\theta_D^{(r_D)},\phi_D^{(q_D)}\rangle\cdots\langle\theta_2^{(r_2)},\phi_2^{(q_2)}\rangle\langle u_1^{(r_1)},u_2^{(q_1)}\rangle\right.\right.$$

$$\left.\left.+\langle\theta_D^{(r_D)},\theta_D^{(q_D)}\rangle\cdots\langle\theta_2^{(r_2)},\theta_2^{(q_2)}\rangle\langle u_1^{(r_1)},u_1^{(q_1)}\rangle+\langle\phi_D^{(r_D)},\phi_D^{(q_D)}\rangle\cdots\langle\phi_2^{(r_2)},\phi_2^{(q_2)}\rangle\langle u_2^{(r_1)},u_2^{(q_1)}\rangle\right)\right)^{1/2}$$

$$=\sigma_{\min}(A)\min_{r_1}\sqrt{\sum_{r_2,\ldots,r_D}G^2(r_1,...,r_D)}, \tag{60}$$

where the last inequality is from the conditions on $\theta_d^{(r)}$ and $\phi_d^{(r)}$. We finish the proof by combining (58), (59), (60), and the fact that

$$\frac{\|\text{vec}(G)\|_1}{\min_{r_1}\sqrt{\sum_{r_2,\ldots,r_D}G^2(r_1,...,r_D)}}\leq CR_1\sqrt{\text{nnz}(G)},$$

where $C=\max_{r_1}\sqrt{\sum_{r_2,\ldots,r_D}G^2(r_1,...,r_D)}/\min_{r_1}\sqrt{\sum_{r_2,\ldots,r_D}G^2(r_1,...,r_D)}$. $\qquad\square$