[Reviews · NeurIPS 2017]

Reviewer 1



This paper studies the problem of sketching the tensor version of the least-squares regression problem. Assuming that the problem is defined by a low-rank tensor it gives a sketching algorithm that reduces the dimension of the problem to rank * (sum of the dimension of factors) instead of the naive bound of the product of dimensions of factors (up to polylog-factors and for constant \epsilon) which roughly matches the number of parameters of the problem. On a technical level this paper introduces new ideas that go far beyond simple generalizations of the work of [3] for OLS regression. Overall, I think this paper is strong enough to be accepted to NIPS and should be of interest to the audience. However, there are some questions I have for the authors (see detailed comments below). Strengths: -- Introduction to the paper is fairly well-written and does a good job introducing technical ideas. -- Interesting problem and a convincing set of theoretical results. The results are clearly stated and the writing is formal throughout the paper. Weaknesses: -- Some of the assumptions made in the main theorem (Theorem 1) were not particularly convincing to me. Also these assumptions come in after the introduction that explicitly stresses that you do not assume anything about the tensor (page 2, line 66). How so? In particular, you justify your first assumption that R <= \sum p_d / 2 by the fact that you are interested in the case R <= p_d. This is the first time the latter assumption seems to appear in the paper and I wasn't sure why is it reasonable. Similar question arises for the second assumption on the leverage scores that you are using. -- When it comes to the technical sections the writing gets patchy. Sometimes the notation gets too heavy and I couldn't follow some of the notation in places (see specific examples below). -- I couldn't quite follow experimental evaluation, starting from the description of the setup. E.g. how was parameter \eps set? Given abundance of log-factors and 1/eps^2 dependence in the sketch dimension I am very surprised to see this approach turns out to be practical. How is this explained? Typos and notation issues: -- line 5: \sum p_D -> \sum p_d -- line 196: in more in -> in more detail in -- line 196: from -> for -- line 198: A^{v_1} -- what does this notation mean -- line 204: \mathcal N -- I assume this is a covering number of some kind (probably L2?), but couldn't find it being defined anywhere

Reviewer 2



*Summary* This paper studies the tensor L2 regression problem and proposed to use matrix sketching to reduce computation. 1+epsilon bound is established to guarantee the error incurred by sketching. I don't like the writing. First, there is little discussion of and comparison with related work. Second, the proof in the main body of the paper is messy, and it takes much space which could have been used for discussion and experiments. Third, the use of notation is confusing. Some claims in the paper seems false or suspicious. I'd like to see the authors' response. I'll adjust my rating accordingly. *Details* I'd like to see the authors' response to Questions 2, 4, 9, 10. 1. I don't find the description of your algorithm anywhere. I don't find the time complexity analysis anywhere. I want to see a comparison of the computational complexities w/ and w/o sketching. 2. How is this paper related to and different from previous work, e.g., [32], Drineas's work on sketched OLS, and tensor sketch? For example, if A is matrix, which is a special case of tensor, then how does your bounds compare to Drineas's and Clarkson&Woodruff's? 3. The notation is not well described. I look back and forth for the definition of D, p, r, R, m, s, etc. 4. In Line 67, it is claimed that no assumption on incoherence is made. However, in Line 169 and 231, there are requirements on the max leverage scores. How do you explain this? 5. There are incompleteness and errors in the citations. * There are numerous works on sketched OLS. These works should be cited telow Eqn (6). * In Definition 5: SJLT is well known as count sketch. Count sketch has been studied in numerous papers, e.g, Charikar et al: Finding frequent items in data streams. Clarkson & Woodruff: Low rank approximation and regression in input sparsity time Meng and Mahoney: Low-distortion subspace embeddings in input- sparsity time and applications to robust linear regression Nelson & Nguyen: Osnap: Faster numerical linear algebra algorithms via sparser subspace embeddings. Pham and Pagh. Fast and scalable polynomial kernels via explicit feature maps Thorup and Zhang. Tabulation-based 5-independent hashing with appli- cations to linear probing and second moment estimation I don't think [11] is relevant. 6. How does this work extend to regularized regression? 7. Line 182: How is the assumption mild? The denominator is actually big. Matrix completion is not a good reference; it's a different problem. 8. The synthetic data in the experiments are not interesting. Sampling from normal distribution ensures incoherence. You'd better generate data from t-distribution according to the paper * Ma et al: A statistical perspective on algorithmic leveraging. I'd like to see real data experiments to demonstrate the usefulness of this work. 9. The title claims "near optimal". Why is this true? Where's your lower bound? 10. Tensors are rarely low-rank; but sometimes they are approximately low-rank. How does your theory and algorithm apply in this case? === after feedback === I appreciate the authors' patient reply. I think my evaluation of the technical quality is fair. I won't change my rating.

Reviewer 3



This paper discusses how sparse JL transform can be constructed and used for tensor regression, where the tensors are vectorized. For notation simplicity, I will assume the tensors are D-way and each mode is of size p, so the vectorized tensors are of dimension {Dp}. My understanding of the author's argument is that using sparse JL, the problem size is reduced where there are only O(R * Dp polylog_factor ) equations in the approximated linear inverse problem, and this matches the problem's intrinsic degrees of freedom. I'm not familiar enough with the sketching literature, to help me understand the paper more in limited time, I'd like to ask a few questions: 1) I think the structure of low rank tensor is considered when proving the bound, but not utilized in the construction of linear system. The objective is squared loss only. I'm wondering if one can sketch a system of smaller size, by imposing some regularizers or constraints for the structure? For example, say D is an even number, using the squared norm [1], we only need n = O(R * p^{D/2}) linear measurement of the tensor. If we sketch this system, by JL theorem we will have m = O(D log (Rp) ) -- which is much smaller. Is this doable? 2) Related to 1), I would appreciate the author discussed the relationships among m (sketched size), n(#measurement) and tensor parameters (R, p, D) more clearly in the paper. Because the JL transform is a relationship between m & n, and the relationship between n & (R, p, D) is more or less the main barrier for tensor recovery problem. It seems these two relationships are mixed in this paper, the authors connects m and (R, p, D) directly. 3) I'm interested in the experimental results for Fig 1(b). I think the Y-axis plots MSE, X-axis plots iteration number. The number of measurement is of order n3 > n2 > n1, but the convergence speed you get is n1 (fastest) > n3 > n2. Could you please explain why the error decreases slower with more data? Thanks! [1] Square Deal: Lower Bounds and Improved Relaxations for Tensor Recovery